# Tamoxifen mechanically reprograms the tumor microenvironment via HIF-1A and reduces cancer cell survival

Ernesto Cortes[1], Dariusz Lachowski[1] [ID], Benjamin Robinson[1], Muge Sarper[1], Jaakko S Teppo[2] [ID], Stephen D Thorpe[3], Tyler J Lieberthal[1], Kazunari Iwamoto[4,5], David A Lee[3] [ID], Mariko Okada-Hatakeyama[4,5], Markku T Varjosalo[2] & Armando E del Río Hernández[1,*] [ID]

## Abstract

The tumor microenvironment is fundamental to cancer progression, and the influence of its mechanical properties is increasingly being appreciated. Tamoxifen has been used for many years to treat estrogen-positive breast cancer. Here we report that tamoxifen regulates the level and activity of collagen cross-linking and degradative enzymes, and hence the organization of the extracellular matrix, via a mechanism involving both the G protein-coupled estrogen receptor (GPER) and hypoxia-inducible factor-1 alpha (HIF-1A). We show that tamoxifen reduces HIF-1A levels by suppressing myosin-dependent contractility and matrix stiffness mechanosensing. Tamoxifen also downregulates hypoxia-regulated genes and increases vascularization in PDAC tissues. Our findings implicate the GPER/HIF-1A axis as a master regulator of peri-tumoral stromal remodeling and the fibrovascular tumor microenvironment and offer a paradigm shift for tamoxifen from a well-established drug in breast cancer hormonal therapy to an alternative candidate for stromal targeting strategies in PDAC and possibly other cancers.

**Keywords** GPER; HIF-1A; tamoxifen; tumor microenvironment

**Subject Categories** Cancer; Signal Transduction

See also: **E Cortes et al** (January 2019) and **M Pein & T Oskarsson** (January 2019)

## Introduction

Pancreatic ductal adenocarcinoma (PDAC) is the most common type of pancreatic cancer and one of the leading causes of cancer-related death despite substantial efforts in recent years aimed at optimizing therapies. PDAC is distinguished by a strong desmoplasia in the tumor microenvironment or stroma that has been associated with aggressiveness of the disease [1], but has also been reported to restrain tumor growth [2,3], suggesting that stromal contribution varies depending on the context. Therefore, there is an urgent need for studies that characterize the structure of this desmoplastic stroma in order to decipher its complex and dynamic interaction with the tumor. PDAC is one of the stiffest solid carcinomas, which intuitively leads to the idea of disrupted mechanical communication between cancer and stromal cells and unbalanced tissue tension within the extracellular matrix (ECM). Thus, PDAC can be viewed as a complex multifaceted disease of altered mechanobiology. Indeed, a recent study has shown that enhanced mechanosignaling in the tumor epithelia can promote PDAC progression in mouse models, overriding the need for p53 mutation [4], while another study showed that targeting focal adhesion kinase *in vivo* could reduce fibrosis and therefore sensitize pancreatic cancer cells to immunotherapy [5].

The strong desmoplasia severely impacts vascular function in PDAC, which hosts a remarkably hypovascularized tumor microenvironment. This dysfunctional vasculature has represented a major hurdle for chemotherapy delivery and has been used as a diagnostic tool in PDAC imaging for many years [1]. PDAC, like other hypovascularized tumors, has substantial hypoxic areas [6,7]. The ability of cancer and stromal cells to thrive under these hostile conditions of subpar oxygen supply depends on their capacity to trigger pathways necessary for development under hypoxic conditions. The hypoxia-inducible factor (HIF) pathway is the main mechanism activated in cells to adapt to hypoxia. Under these conditions, hypoxia-inducible factor-1 alpha (HIF-1A) translocates to the nucleus and binds to the hypoxia-response elements, thereby activating the expression of genes that control multiple functions in cells such as metabolism, survival, proliferation and apoptosis, migration, energetic balance, and pH [8]. Notably, PDAC seems to progress without the need of excessive angiogenesis and a recent

1 Cellular and Molecular Biomechanics Laboratory, Department of Bioengineering, Imperial College London, London, UK
2 Institute of Biotechnology, University of Helsinki, Helsinki, Finland
3 Institute of Bioengineering, School of Engineering and Materials Science, Queen Mary University of London, London, UK
4 Laboratory of Cell Systems, Institute for Protein Research, Osaka University, Suita, Osaka, Japan
5 Laboratory for Integrated Cellular Systems, RIKEN Center for Integrative Medical Sciences (IMS), Yokohama, Kanagawa, Japan
   *Corresponding author. Tel: +44(0)2075948157; E-mail: a.del-rio-hernandez@imperial.ac.uk

study suggests a lack of correlation between the hypovascular nature of PDAC and hypoxia [9,10].

Pancreatic stellate cells (PSCs) are the main group of resident cells in the stroma and the key drivers of the desmoplastic reaction [11]. In PDAC, like other cancer-associated fibroblasts (CAFs), PSCs are activated and adopt a myofibroblastic phenotype with high contractile activity, leading to stiffening of the ECM and extensive deposition of ECM proteins such as collagen and fibronectin [12–14]. PSCs orchestrate ECM organization, not only via force-mediated matrix remodeling or through the synthesis and deposition of ECM proteins, but also by regulated secretion of elevated levels of matrix cross-linking enzymes such as lysyl oxidase (LOX) and degradative proteases such as metalloproteinases (MMPs) and their inhibitors (tissue inhibitor of metalloproteinases, TIMPs) [11,15,16]. The controlled balance between these cross-linking and degradative enzymes regulates ECM architecture in normal pancreas, but loss of this balance in PDAC triggers and sustains the desmoplastic reaction [12]. Interestingly, the LOX/hypoxia axis correlates with poor prognosis in PDAC patients and targeting this axis in PDAC mice has been shown to decrease tumorigenesis, augment chemotherapy efficacy, and decrease metastasis [17]. Moreover, treating PDAC mouse models with ATRA (all trans-retinoic acid), which abrogates force-mediated ECM remodeling by PSCs [16], increased vascular density and decreased hypoxia [18].

Tamoxifen has been used for many years to treat breast cancers based on its genomic effect on the nuclear estrogen receptors. Here we report a previously unidentified mechanism that is independent of the nuclear estrogen receptors and involves the G protein-coupled estrogen receptor (GPER) and hypoxia-inducible factor-1 alpha (HIF-1A). We show that tamoxifen reduces the adaptive response of PDAC to hypoxia via a mechanical downregulation of HIF-1A, and increases vascularization in PDAC mouse models. Tamoxifen tunes the balance between cross-linking (LOX) and degrading enzymes (MMPs) secreted by PSCs to modulate collagen and fibronectin fiber architecture in the tumor microenvironment. Tamoxifen treatment also decreases the fitness of pancreatic cancer cells to cope with hypoxic conditions via mechanical downregulation of HIF-1A.

# Results

## Tamoxifen treatment induces changes in protein content of PDAC tissues and gene expression profiles in PSCs

We treated KPC mice (Pdx-1 Cre, KRas$^{G12D/+}$, p53$^{R172H/+}$), which are known to recapitulate the clinical and histological features of human PDAC [2], with 2 mg of tamoxifen, and used quantitative shotgun proteomic analysis to investigate whether tamoxifen treatment induced changes in the protein content of PDAC tissues. This dose in mice (100 mg/kg) produces a tamoxifen serum concentration around 0.5 μM, which corresponds to the serum concentration found in humans after administration of clinical doses of 20 mg/day [19]. In total, 110 proteins showed statistically significant changes (Fig EV1 and Dataset EV1–EV3). From this group, more than half of the downregulated proteins are involved in ECM organization, cell adhesion, and wound healing. These data have been deposited in the PeptideAtlas under the reference PASS01070.

To study the effect of tamoxifen on the main resident cells in the tumor microenvironment, PSCs were treated with 5 μM of tamoxifen as higher doses showed toxicity (Fig EV2). RNA sequencing and gene profile analysis of control and treated PSCs showed that from the nearly 20,000 expressed genes, 649 were upregulated and 688 downregulated (Fig EV3, Dataset EV4–EV6). The larger group of upregulated genes is associated with blood vessel morphogenesis, and the downregulated genes are involved in ECM organization, cell migration, cell–ECM adhesion, and the response to hypoxia. These data have been deposited in the European Nucleotide Archive, accession number ERP023834.

## Tamoxifen reduces hypoxia and increases vascularization in PDAC tissues

In order to investigate the effect of tamoxifen treatment on hypoxia and vascularization levels in PDAC tissues, we used GLUT1 and CD31 immunofluorescence staining of pancreatic tissues from KPC mice treated with 2 and 5 mg of tamoxifen (Fig 1A). The level of the hypoxia marker GLUT1 was significantly reduced from control mice to mice treated with 5 mg (fourfold reduction) in a dose-dependent fashion (Fig 1B). We also observed a pronounced reduction in Glut1 content in PDAC tissues from treated mice relative to control using quantitative proteomics analysis (Fig 1C). A twofold increase in vascularization was observed in mice treated with the highest doses compared to untreated mice (Fig 1D).

RNA sequencing of PSCs revealed more than 25 hypoxia-related genes downregulated and more than 30 blood vessel morphogenesis genes upregulated after tamoxifen treatment (Figs 1E and F, and EV3). We then focused on the hypoxia-inducible factor (HIF) and vascular endothelial growth factor (VEGF) families as key players associated with hypoxia and blood vessels respectively (Appendix Fig S1). For the HIF family, we found that while HIF-3A did not change, HIF-1A was significantly reduced and HIF-2A (also known as EPAS1) was significantly upregulated. We observed an increase in the expression levels of VEGFB and a downregulation of VEGFC. qPCR was used to validate RNA sequencing data that showed a clear trend but did not display significant differences.

The analysis of the downregulated hypoxia-related genes revealed a map of interactions centered in HIF-1A (Appendix Fig S2). We observed a nearly 70% reduction in HIF-1A levels in PDAC tissues from KPC mice treated with 5 mg of tamoxifen relative to control KPC mice treated with vehicle (Fig 1G and H). We also confirmed that the effect of tamoxifen on the HIF-1A levels of PSCs is mediated by GPER, as the use of GPER antagonist (but not ER antagonist) returned the HIF-1A values to the levels found in control PSCs (Fig 1I). We used immunoblotting to investigate the effect of tamoxifen on the levels of HIF-1A in PSCs and observed an overall 25% reduction in the 3 main HIF-1A isoforms (1–3) and also in the post-translationally modified HIF-1A (Fig 1J and Appendix Fig S3). Taken together, these data show that tamoxifen induces the expression of genes that promote blood vessels formation and downregulates hypoxia-related genes, with many of them converging in HIF-1A.

## Tamoxifen decreases LOX-L2 levels in PSCs and PDAC tissues

The lysyl oxidase (LOX) gene encodes the lysyl oxidase family of extracellular copper-dependent enzymes that catalyses the

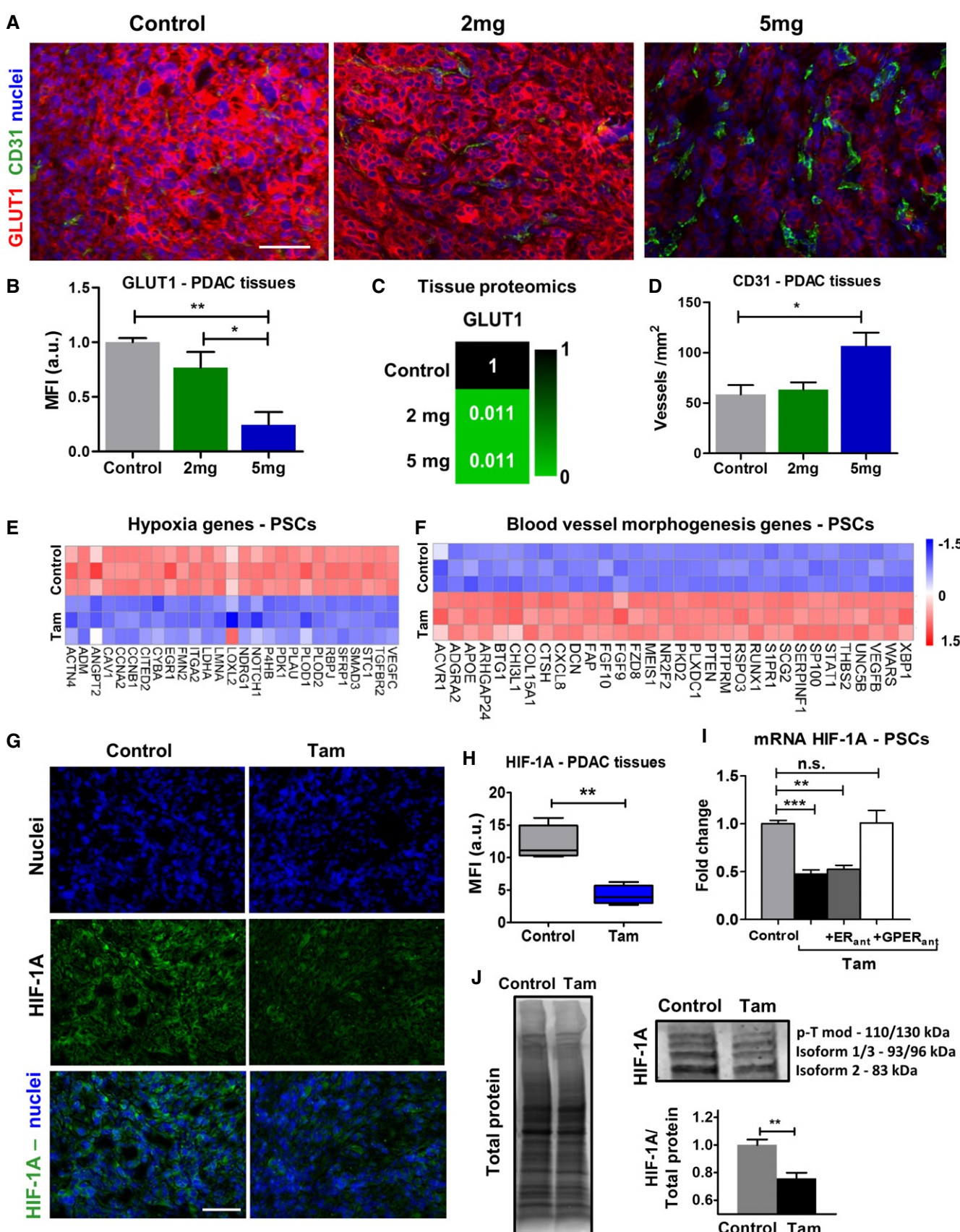

**Figure 1.**

**Figure 1.  Tamoxifen decreases hypoxia and increases vascularization.**

A       Immunofluorescence images of PDAC tissues from KPC mice treated with vehicle control of tamoxifen, scale bar 100 μm.

B–D    (B, D) Quantification of GLUT1 (hypoxia marker) and CD31 (endothelial cell marker). Control (n = 5), 2 mg (n = 5), and 5 mg (n = 4). In all cases, bars represent
        mean ± SEM. (C) Relative values of protein levels for Glut1 in PDAC tumors assessed by proteomic analysis (6 mice for control and 2 mg and 3 mice for 5 mg, and
        samples were analyzed in duplicates).

E, F    Expression levels of DEGs relevant to response to hypoxia (left) and blood vessel morphogenesis (right). The values were normalized by tubulin family genes.

G       Immunofluorescence images of PDAC tissues from KPC mice treated with vehicle control and 5 mg of tamoxifen, scale bar 100 μm.

H       Quantification of HIF-1A in PDAC tissues. Control (n = 5) and 5 mg (n = 4). In the box-and-whisker plot, the central box represents values from the lower to upper
        quartile. The middle line represents the mean. The vertical line extends from the minimum to the maximum value.

I       qPCR levels of HIF-1A in PSCs, normalized to RPLP0 and relative to control.

J       Western blot bands for protein expression in PSCs (p-Tmod is post-translational modification). The plot shows the quantification of the sum of band intensities
        corresponding to isoform 1, isoform 2, isoform 3, and post-transcriptionally modified HIF-1A (n = 8 control and n = 8 tam).

Data information: All histogram bars represent mean ± SEM; *$P < 0.05$, **$P < 0.01$, ***$P < 0.001$ (*t*-test for H and J; ANOVA and Tukey's *post hoc* test for B, D, I). For (A
and G), $n \geq 5$ sections per animal. Results collected during 3 or more separate experiments.

cross-linking of collagen fibers. Within this family, LOX is the most characterized member and LOX-L2 (lysyl oxidase homolog-2) has been comprehensively documented to participate in ECM remodeling of the tumor and fibrotic microenvironments. All LOX members contain a highly conserved copper binding and catalytic C-terminal domain, responsible for the cross-linking function of these enzymes [20]. The LOX family has been known to promote fibrosis and tumorigenesis, and accumulated evidence supports the use of β-aminopropionitrile (βAPN) and simtuzumab to inhibit LOX and LOX-L2 activities in fibrosis and cancer [13]. These inhibitors have reverberated across the fields of inflammation and cancer as potential agents to restore normal collagen architecture and mechanical tissue homeostasis [21].

Increased expression of LOX-L2 has been reported in human PDAC tissues with respect to normal stroma, and elevated LOX/hypoxia is associated with the shortest patient survival [17,22]. Given that we observed a significant downregulation of LOX-L2 after tamoxifen treatment in the analysis of the gene profile of PSCs, and LOX-L2 is elevated in PDAC and regulated by HIF-1A [23,24], we focused first on this member of the LOX family and tested the LOX-L2 levels in PSCs following tamoxifen treatment. We observed a threefold decrease of LOX-L2 at the gene and protein levels compared to untreated control (Fig 2A and B). This effect was maintained when we used an estrogen receptor (ER) antagonist but not a GPER antagonist, which supports the notion that tamoxifen reduces LOX-L2 expression via GPER.

Next, we examined the effect of tamoxifen treatment on the expression levels of all LOX family members in PSCs and PDAC tissues from KPC mice using RNA sequencing, qPCR, and quantitative proteomics analysis. Except for LOX-L1, we observed a marked decrease in the mRNA levels of all members expressed by PSCs, and there was a significant downregulation in the protein content of LOX-L2 and LOX-L3 in tissues coming from KPC mice treated with 5 mg tamoxifen (Fig 2C–E). This dose-dependent significant decrease of LOX-L2 in tissues was also validated by dual staining of LOX-L2 and αSMA (marker of PSCs) immunofluorescence analysis (Fig 2F and G).

In order to gain insights in the mechanism by which tamoxifen decreases LOX-L2 levels in PSCs, we cultured these cells in polyacrylamide (PAA) matrices of varying rigidities: 1 kPa (soft matrix) or 25 kPa (stiff matrix) as previously reported [25,26]. LOX-L2 and HIF-1A levels were significantly upregulated in PSCs cultured on stiff compared to the soft matrices, and tamoxifen-treated PSCs

cultured on stiff substrates expressed LOX-L2 and HIF-1A levels comparable to the ones shown on the soft matrices (Fig 2H). These results led us to investigate whether increased contractility in PSCs could per se increase the levels of expression of LOX-L2 and HIF-1A. We transfected PSCs with a myosin isoform that was constitutively active and monitored the levels of LOX-L2 and HIF-1A from PSCs cultured on glass. Both LOX-L2 and HIF-1A were significantly upregulated in PSCs transfected with active myosin compared to the values from control PSCs (Fig 2I). We used micropillars sensors as described previously [16] to observe that tamoxifen decreased myosin-dependent contractility in PSCs, with the endogenous forces applied to the matrix by tamoxifen-treated PSCs comparable to the forces observed with the use of blebbistatin (BBI), a strong inhibitor of myosin activation and cell contractility (Fig 2J). This decrease in traction forces was maintained with tamoxifen treatment in the presence of the ER antagonist but suppressed when the GPER antagonist was used. We also used siRNA against HIF-1A to knockdown HIF-1A expression in PSCs (Appendix Fig S4) and found that under these conditions, tamoxifen treatment did not reduce LOX-L2 levels beyond the values observed for PSCs transfected with siRNA against HIF-1A (Fig 2K). Collectively, these data show that tamoxifen decreases LOX-L2 in PSCs and PDAC tissues via GPER signaling and through a mechanism that involves mechanical downregulation of HIF-1A via myosin-dependent PSC contractility and matrix stiffness mechanosensing.

## Tamoxifen modulates collagen synthesis and matrix collagen remodeling by PSCs

In order to study the effect of tamoxifen on the ability of PSCs to remodel the matrix, we used 3D organotypic assays in which PSCs were embedded within matrigel collagen gels and allowed to remodel the matrix for 72 h. Then, second harmonic generation (SHG) imaging was used to visualize type I fibrillar collagen (referred to as collagen hereafter) (Fig 3A). Untreated control PSCs modified the topology of fibrillar collagen in the ECM substantially differently than their tamoxifen-treated counterparts. We used the BoneJ plugin for ImageJ to quantify the thickness of fibrillar collagen fibers (Fig 3B), and a custom-made algorithm based on fast Fourier transforms (FFT) [15] to quantify the elliptical distribution of the fibrillar collagen network that indicates fiber alignment (Fig 3C). In this analysis, highly aligned collagen fibers exhibit values closer to 0 (elliptical distribution in Fig 3C inset), while

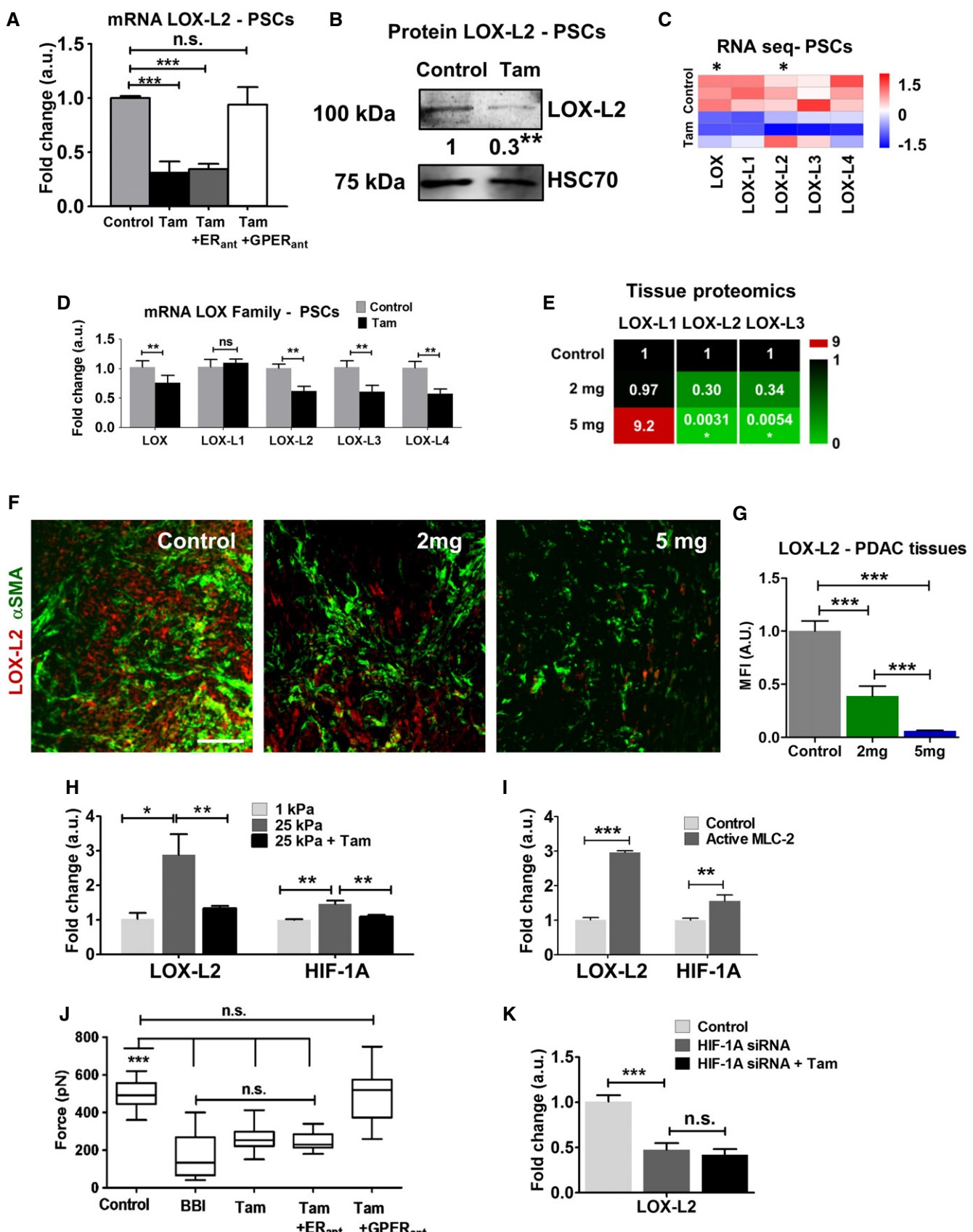

Figure 2.

**Figure 2. Tamoxifen reduces LOX-L2 levels in PSCs and PDAC tissues.**

A   qPCR levels of LOX-L2 in PSCs, normalized to RPLP0 and relative to control.
B   Western blot levels of LOX-L2 in PSCs (*n* = 3 experimental replicates).
C   Expression of LOX family genes obtained from RNA-seq data in control and tamoxifen-treated PSCs (*n* = 3 experimental replicates). Expression value was normalized by tubulin family genes. Asterisk means significant differences (*P* < 0.05). Mann-Whitney *U*-test.
D   qPCR levels of LOX family in PSCs, normalized to RPLP0 and relative to control.
E   Relative values of protein levels for LOX members in PDAC tumors assessed by proteomic analysis (6 mice for control and 2 mg and 3 mice for 5 mg, and samples were analyzed in duplicates).
F   Immunofluorescence images of PDAC tissues from KPC mice treated with vehicle (control), and 2 mg and 5 mg of tamoxifen, scale bar 50 μm.
G   Quantification of LOX-L2 for images in (F). *n* = 5 (control), 4 (2 mg), and 3 (5 mg), and *n* > 10 sections per animal.
H   qPCR levels of LOX-L2 and HIF-1A in PSCs, normalized to RPLP0 and relative to 1 kPa.
I   qPCR levels of LOX-L2 and HIF-1A in PSCs, normalized to RPLP0 and relative to control.
J   Quantification of average forces applied by PSCs on pillars. BBI = blebbistatin. In the box-and-whisker plot, the central box represents values from the lower and upper quartile. The middle line represents the mean. The vertical line extends from the minimum to the maximum. Three experimental repeats.
K   qPCR levels of LOX-L2 in PSCs, normalized to RPLP0 (60S acidic ribosomal protein P0) and relative to control.

Data information: In all cases, histogram bars represent mean ± SEM. *P < 0.05, **P < 0.01, ***P < 0.001 (*t*-test for B, D, I; ANOVA and Tukey's *post hoc* test for A, G, H, J, K). For (A, D, H, I, J, and K) three replicates collected in at least three different experiments.

randomly aligned fibers values close to 1 (circular distribution in Fig 3C inset). Overall, ECM remodeled by tamoxifen-treated PSCs showed a significant decrease in fibrillar collagen fiber diameter, length, and alignment compared to control PSCs (Fig 3D–F).

LOX-L2 catalyses the first step in the cross-linking of collagen bundles. In order to causally relate the decrease in the diameter of collagen fibers under tamoxifen treatment with LOX-L2 activity, we treated PSCs with tamoxifen, rescued the treated cells with LOX-L2, and allowed PSCs to remodel the collagen matrigel matrices in the same manner as previously explained. Analysis of the SHG images revealed that the diameter of the collagen fibers in matrices remodeled by LOX-L2 rescued tamoxifen-treated PSCs was not significantly different than the diameter of the fibers in matrices remodeled by untreated control PSCs (Fig 3G and H). Furthermore, a LOX-L2-specific antibody returned the diameter of the fiber in the matrices remodeled by tamoxifen treatment plus LOX-L2 rescued PSCs to statistically comparable values of the diameter in matrices remodeled by tamoxifen-treated PSCs alone.

Furthermore, the amount of collagen produced by PSCs was downregulated at the gene and protein levels, as well as collagen secretion (Fig 4A–D). This effect was maintained in the presence of ER antagonist but abrogated when GPER antagonist was used. We also observed a consistent dose-dependent downregulation of collagen in PDAC tissues from KPC mice treated with tamoxifen (Fig 4E). Collectively, these data show that tamoxifen reduces collagen synthesis via GPER signaling and collagen remodeling via LOX-L2.

## Tamoxifen downregulates MMP-2 in PSCs and PDAC tissues

Matrix metalloproteinases (MMPs, also known as matrixins) are considered the main group of ECM degrading enzymes [27]. MMPs are a family of calcium-dependent zinc-containing endopeptidases, of which MMP-2 and MMP-9 are the two main types secreted by PSCs [28]. It has been reported that HIF-1A regulates the expression of the MMP-2 gene [29]. After observing a downregulation of HIF-1A levels in tamoxifen-treated PSCs, we investigated whether tamoxifen interfered with the proteolytic remodeling ability of PSCs through MMP-2. Tamoxifen treatment significantly downregulated MMP-2 expression and activity in a GPER-dependent manner, assessed by PCR and zymography (Fig 4F and G). MMPs are closely regulated by specific endogenous tissue inhibitors of metalloproteinases (TIMPs), and TIMP-2 is a specific inhibitor of MMP-2 [30]. In order to test whether the changes in MMP-2 levels in PSCs under tamoxifen treatment might be influenced by TIMP-2, we determined the levels of expression of TIMP-2 in PSCs and observed that there were no significant changes in TIMP-2 mRNA levels in tamoxifen-treated PSCs in relation to untreated control PSCs (Appendix Fig S5). We used immunohistochemistry and quantitative proteomics to quantify the MMP-2 levels in tissues from KPC mice treated with 2 and 5 mg of tamoxifen. MMP-2 mRNA levels were significantly decreased in a dose-dependent manner from control mice to mice treated with 5 mg of tamoxifen (Fig 4H–J).

In order to learn more about the mechanism by which tamoxifen decreased MMP-2 levels in PSCs, we cultured PSCs on 1 and 25 kPa PAA matrices and observed a significant increase in MMP-2 values

**Figure 3. Tamoxifen decreases collagen fiber thickness, length, and alignment.**

A   Images of Matrigel collagen gels previously remodeled by PSCs, second harmonic generation signal for fibrillar collagen (green) and F-actin (red), scale bar 100 μm.
B   Fiber thickness color-code map in a represented through the BoneJ plugin where larger spheres fit along fibers represent greater thickness, scale bar 100 μm.
C   SHG fibrillar collagen images used for calculation of alignment through the FFT algorithm. Insets show FFTs of fibrillar collagen-I images, representing alignment with respect to the elliptical distribution of the FFT central maxima. Circular behavior (values approaching 1) represents no aligned orientation and lower values represent fiber orientation as alignment is displayed as a power distribution orthogonal to the orientation direction. Scale bar 20 μm.
D–F Quantification of fiber thickness, length, and alignment for images in (A–C).
G   Quantification of collagen fiber thickness using the BoneJ algorithm for images in (B and H).
H   Representative images of Matrigel collagen gels previously remodeled by PSCs, second harmonic generation signal for fibrillar collagen (green) and F-actin (red), scale bar 100 μm.

Data information: *n* ≥ 6 matrices per condition. In the scatter plot in (E), each point represents a section. In the box-and-whisker plot in (D), the central box represents values from the lower to upper quartile. The middle line represents the mean. The vertical line extends from the minimum to the maximum value. Histogram bars (F and G) represent mean ± SEM. ***P < 0.001 (*t*-test for D, E, F; ANOVA and Tukey's *post hoc* test for G). Three experimental replicates for all panels.

on the stiffest matrices. MMP-2 values were significantly reduced in tamoxifen-treated PSCs cultured in 25 kPa (Fig 4K). We used siRNA against HIF-1A to knockdown HIF-1A expression in PSCs and found that under these conditions, tamoxifen treatment did not reduce MMP-2 levels beyond the values observed for PSCs transfected with siRNA against HIF-1A (Fig 4L). We used qPCR and zymography to

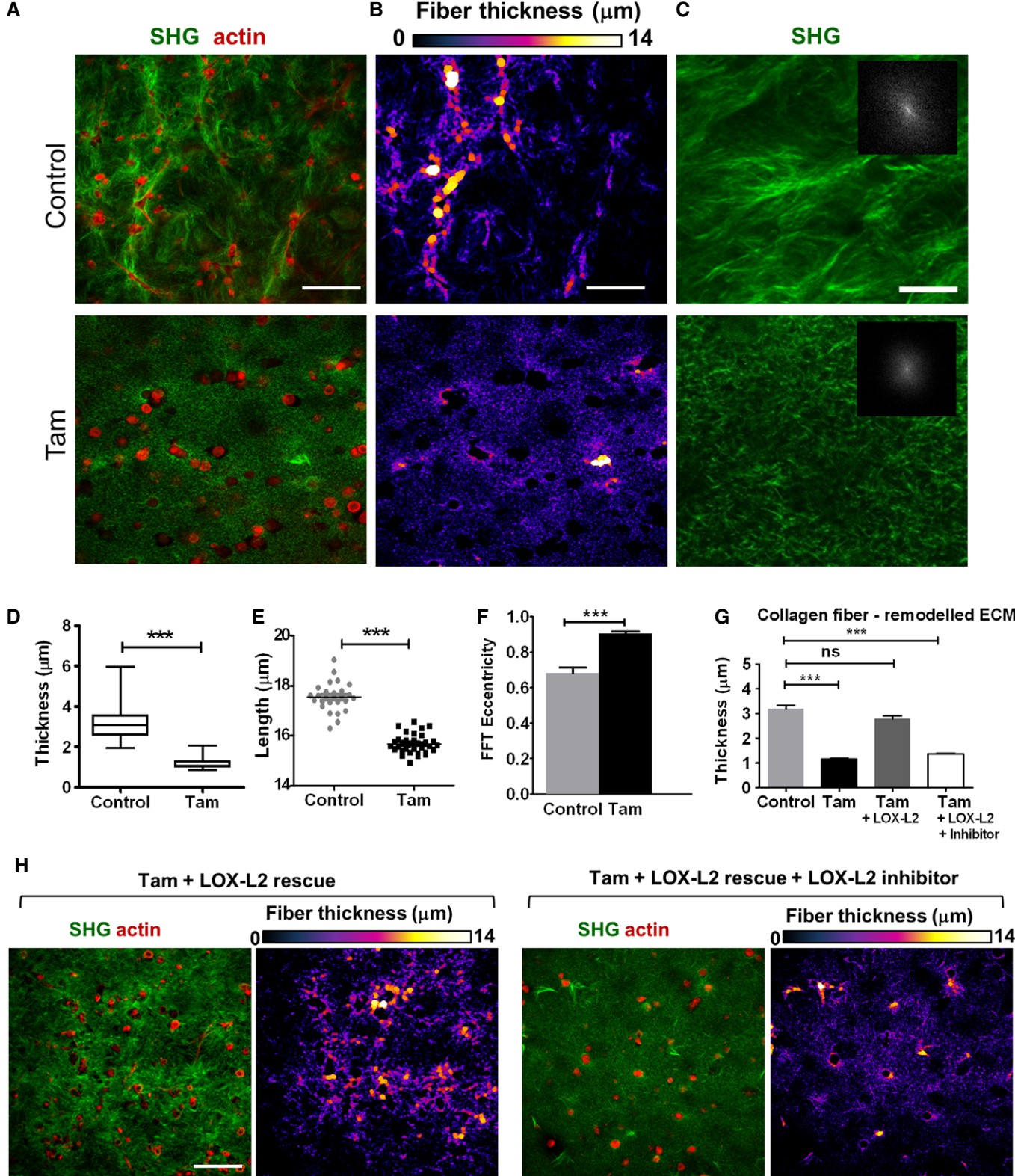

**Figure 3.**

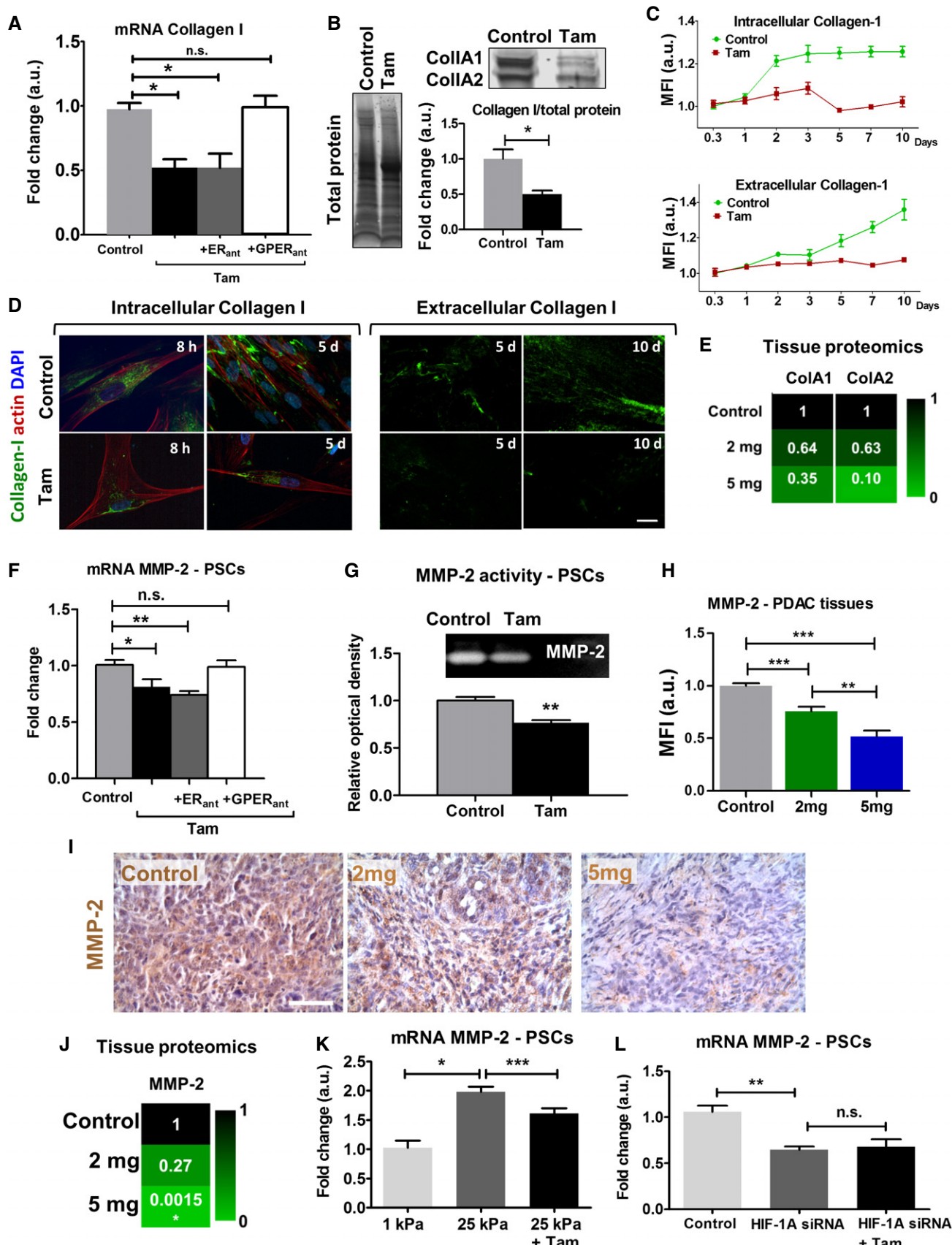

Figure 4.

**Figure 4.  Tamoxifen decreases synthesis and secretion of collagen and MMP-2 in PSCs and PDAC tissues.**

A    qPCR levels of collagen in PSCs, normalized to RPLP0 and relative to control.
B    Western blot analysis of collagen normalized to total protein and relative to control.
C    Quantification of time-lapse collagen synthesis and deposition by PSCs.
D    Representative immunofluorescent images used for the quantification in (C), scale bar 20 μm, collagen-I was assessed with a specific primary antibody staining.
E    Relative values of protein levels for collagen in PDAC tumors assessed by proteomic analysis (6 mice for control and 2 mg and 3 mice for 5 mg, and samples were analyzed in duplicates).
F    qPCR levels of MMP-2 in PSCs, normalized to RPLP0 and relative to control.
G    MMP-2 activity on control and tamoxifen-treated PSCs assayed by gelatin zymography; above signal intensity of the representative bands used for the quantification presented in the plot below.
H, I    Immunohistochemistry images and quantification of MMP-2 levels in PDAC tissues from KPC mice treated with vehicle (control), and 2 and 5 mg of tamoxifen, scale bar 100 μm (*n* = 5 (control), 4 (2 mg), and 3 (5 mg) and *n* ≥ 5 sections per animal).
J    Relative values of protein levels for MMP-2 in PDAC tumors assessed by proteomic analysis (6 mice for control and 2 mg and 3 mice for 5 mg, and samples were analyzed in duplicates).
K    qPCR levels of MMP-2 in PSCs, normalized to RPLP0 and relative to 1 kPa.
L    qPCR levels of MMP-2 in PSCs, normalized to RPLP0 and relative to control.

Data information: Histogram bars represent mean ± SEM. *$P < 0.05$, **$P < 0.01$, ***$P < 0.001$ (*t*-test for group of two data sets in B and G and ANOVA and Tukey's *post hoc* test for the rest). For (A, B, C, F, G, H, K, and L) 3 replicates collected in three different experiments.

test the gene expression and enzymatic activity of MMP-9 produced by PSCs, as well as immunofluorescence and quantitative proteomics of PDAC tissues from control and treated KPC mice to validate these findings in tissues. We observed no significant effect of tamoxifen in MMP-9 values (Appendix Fig S5). Taken together, these results suggest that tamoxifen decreases MMP-2 via mechanical downregulation of HIF-1A.

**Fibronectin secretion by PSCs and its organization in the ECM is modulated by tamoxifen**

Together with collagen, fibronectin is the other major ECM molecule secreted by activated PSCs into PDAC stromal tissues [12]. Fibronectin is associated with LOX activity, and its expression is regulated by HIF-1A [29,31]. After observing a decrease in both HIF-1A and LOX-L2 expression with tamoxifen treatment, we sought to study whether fibronectin expression levels were affected following tamoxifen treatment. The fibronectin gene in cancer is regulated through alternative splicing, implying a significant role of the post-transcriptional regulatory mechanism for its differential expression and function in distinct tissue types [32]. For instance, fibronectin extra domain A (EDA) isoform is expressed in liver and breast cancer [33], while fibronectin extra domain B (EDB) has been implicated in the PDAC microenvironment [34]. Both isoforms, EDA and EDB, are secreted by PSCs [35].

We observed a significant decrease in the levels of gene expression of fibronectin (FN) and both splicing variants (FN-EDA and FN-EDB), and also a downregulation of FN at the protein level when PSCs were treated with tamoxifen versus untreated controls cells (Fig 5A and B). Likewise, when PSCs were embedded within collagen matrigel matrices and allowed to deposit fibronectin into the ECM for 72 h, tamoxifen-treated PSCs deposited about half the amount of fibronectin deposited by untreated control PSCs (Fig 5C and D). Similar to what we observed for LOX-L2 and MMP-2, PSCs cultured on 25 kPa matrices produced more FN than PSCs cultured on 1 kPa, and tamoxifen-treated PSCs cultured on 25 kPa produced FN levels comparable to those of PSCs on 1 kPa. We used siRNA against HIF-1A to knockdown HIF-1A expression in PSCs and found that under these conditions, tamoxifen treatment did not reduce FN levels beyond the values observed for PSCs transfected with siRNA against HIF-1A (Appendix Fig S6). This supports the notion of mechanical regulation of FN expression by PSCs via HIF-1A.

We then sought to investigate whether the effect of tamoxifen on PSCs is also observed in PDAC tissues from KPC mice treated with 2 and 5 mg of tamoxifen using immunostaining and quantitative proteomics. We observed a significant and pronounced decrease of FN content in a dose-dependent manner (Fig 5E, G and H). Using the BoneJ plugin for ImageJ and custom-made algorithms, we found that the thickness and alignment of the fibronectin fibers was significantly reduced from control untreated mice to mice treated with 5 mg of tamoxifen (Fig 5F, I and J). Collectively, these data suggest that tamoxifen reduces FN secretion in PSCs and PDAC tissues via mechanical downregulation of HIF-1A.

**Figure 5.  Tamoxifen decreases fibronectin levels in PSCs and PDAC tissues.**

A    qPCR levels of fibronectin (FN), fibronectin extracellular domain A (FN-EDA), and fibronectin extracellular domain B (FN-EDB) in PSCs, normalized to RPLP0 and relative to control.
B    Western blot analysis of FN normalized to total protein and relative to control.
C    Quantification of fibronectin intensity density of images presented in (D).
D    FN fluorescence and SHG collagen images for matrigel collagen gels remodeled by PSCs.
E    Immunohistochemistry images of PDAC tissues from KPC mice treated with vehicle control, and 2 mg, and 5 mg of tamoxifen (*n* = 5 (control), 4 (2 mg), and 3 (5 mg), and *n* ≥ 5 sections per animal).
F    FN fiber thickness color-code map in (E) represented through the BoneJ plugin.
G    Relative values of protein levels for FN in PDAC tumors assessed by proteomic analysis.
H–J    Quantification of fibronectin immunohistochemistry staining, fiber thickness, and alignment scored of images presented in (E).

Data information: In all cases, histogram bars represent mean ± SEM; *$P < 0.05$, **$P < 0.01$, ***$P < 0.001$ (*t*-test for A, B, C; ANOVA and Tukey's *post hoc* test for H, I, J). For (A, B, C, H, I, and J) three replicates collected in three different experiments. Scale bars are 100 μm.

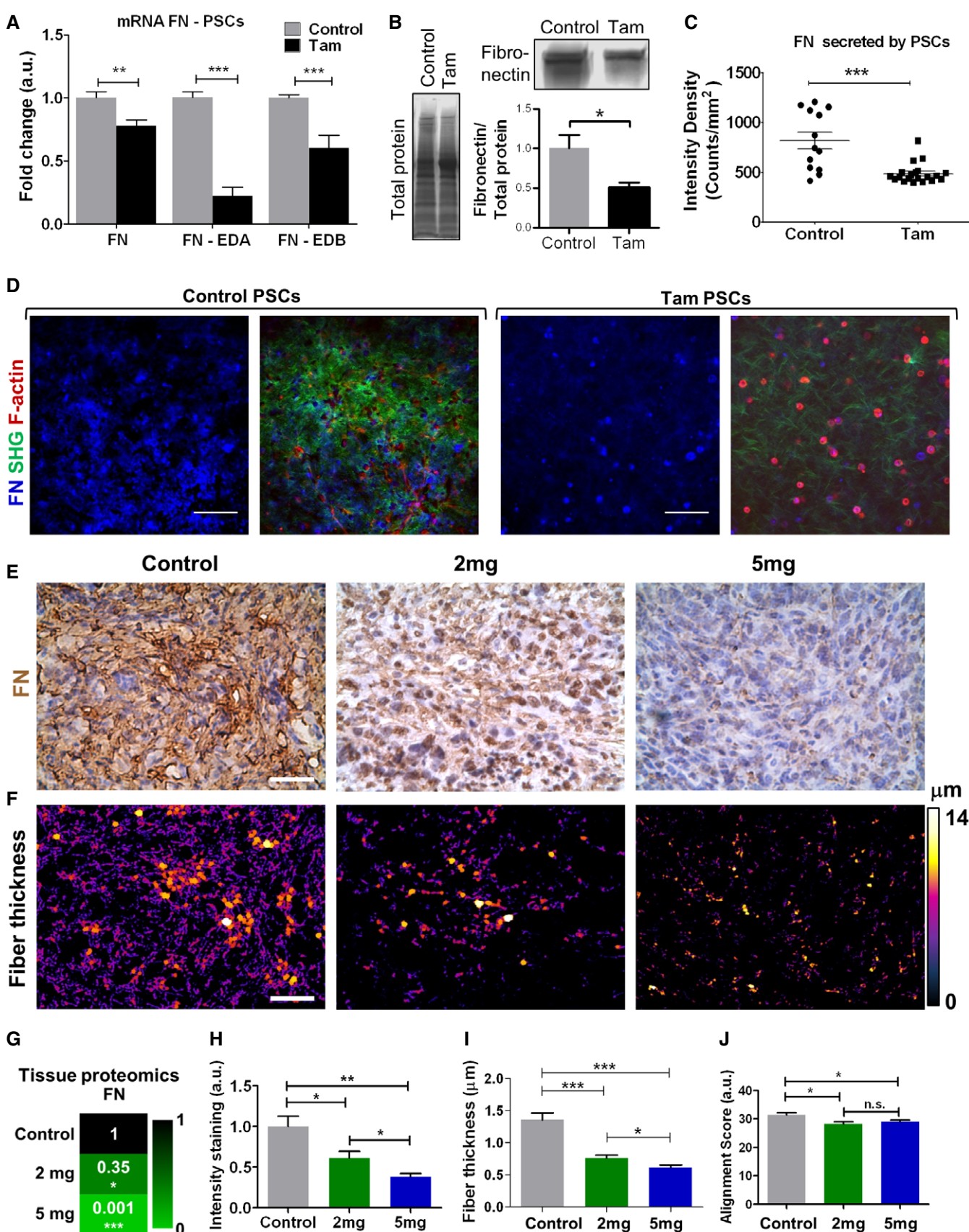

**Figure 5.**

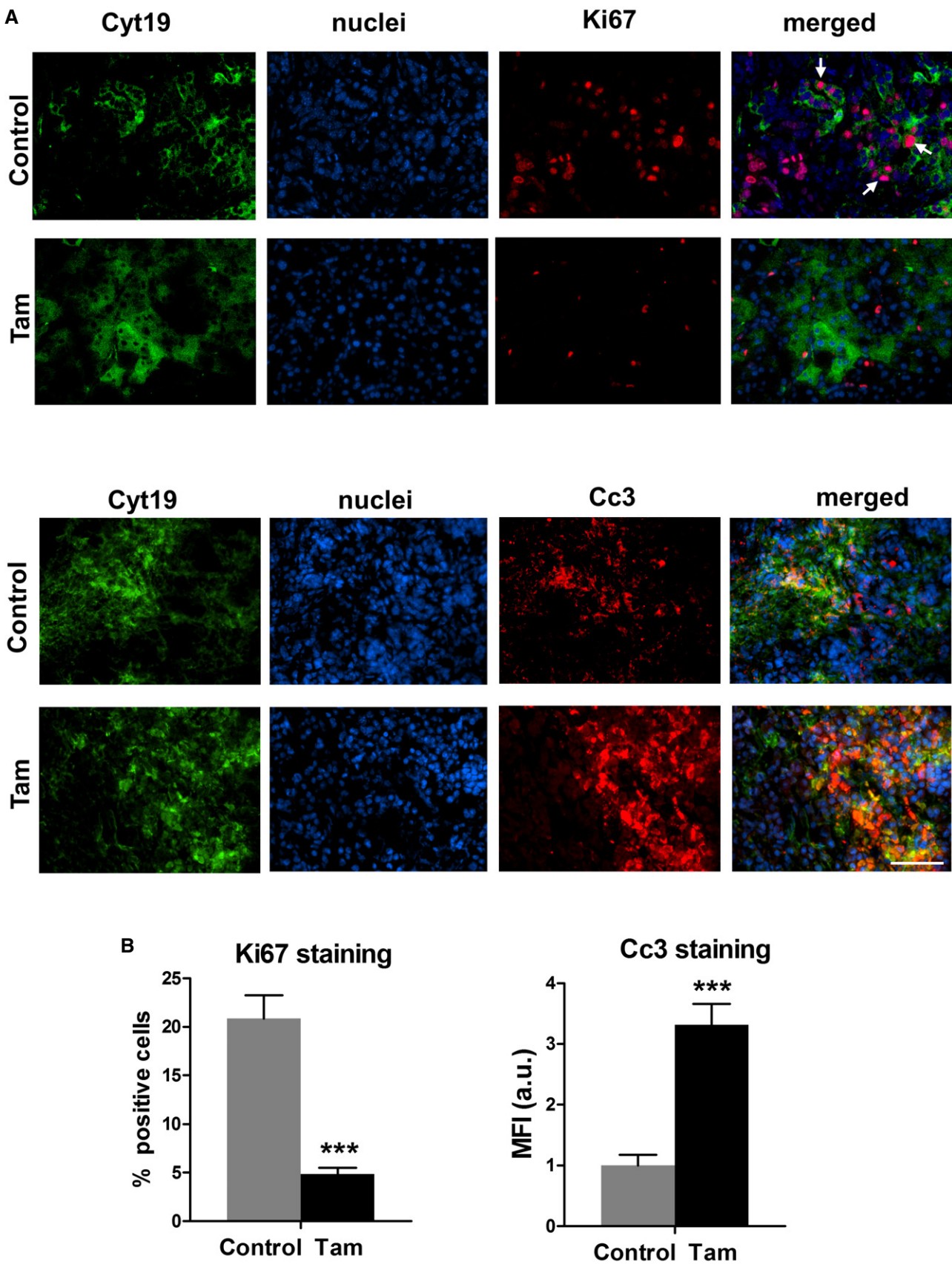

**Figure 6.**

◄

**Figure 6.  Tamoxifen treatment decreases proliferation and increases apoptosis in epithelial cells of PDAC tissues.**

A    Immunofluorescence images of PDAC tissues from KPC mice treated with vehicle control, and 2 mg of tamoxifen, scale bar is 100 μm. Upper panels: Ki67 staining is used as a surrogate of proliferation. White arrows indicate Ki67-positive nuclei in epithelial cells. Lower panels: Cc3 staining shows the cells undergoing caspase-3-mediated apoptosis. Tamoxifen panels show higher levels of yellow staining, which indicates higher percentage of apoptotic epithelial cells.

B    Quantification of staining in panel (A) ($n = 4$ animals per condition, and $n \geq 5$ sections per animal, two experimental repetitions). Histogram bars represent mean $\pm$ SEM; ***$P < 0.001$, $t$-test.

## Tamoxifen treatment decreases proliferation and increases apoptosis in cancer cells

Cancer cells trigger HIF-1A signaling upon hypoxia conditions [8]. Having observed the hypoxia-independent mechanical regulation of HIF-A signaling in PSCs, we sought to investigate whether the survival of cancer cells could be affected by this regulation as well. We used immunofluorescence of tissues from control and tamoxifen-treated KPC mice in which cytokeratin 19 was utilized as a marker for epithelial cells. We observed a significant and marked downregulation of proliferation measured through Ki67 staining and a little over three times increase in the levels of caspase-3-mediated apoptosis (Fig 6A and B) in the cancer cells in tissues from tamoxifen-treated KPC mice.

Next, we examined the *in vitro* effect of tamoxifen treatment on the levels of HIF-1A and the rates of proliferation and apoptosis in the pancreatic cancer cell line Suit2-007 under hypoxic and non-hypoxic conditions. The levels of HIF-1A were significantly reduced in both conditions indicating that tamoxifen can negatively regulate HIF-1A levels in Suit-2 cells thorough a mechanism that is independent of hypoxia (Fig 7A and B). Less surprisingly, the proliferation rate significantly decreased and apoptosis significantly increased in both conditions (hypoxia and non-hypoxia) (Fig 7C–F).

To gain more insights in the mechanism that regulates HIF-1A levels in these cancer cells, we cultured Suit-2 in the soft (1 kPa) and stiff (25 kPa) PAA matrices under non-hypoxic conditions and observed that stiff substrates promoted transcriptional upregulation of HIF-1A. Tamoxifen treatment reduced HIF-1A to levels comparable to the ones shown in the soft matrices (Appendix Fig S7). We also observed that Suit-2 cells expressing the constitutively active isoform of myosin expressed significantly higher levels of HIF-1A compared to control. Treating Suit-2 cells with blebbistatin, an inhibitor of myosin activation significantly reduced the levels of HIF-A compared to control (Appendix Fig S7). These results support the notion that HIF-1A levels in Suit-2 cells are mechanically regulated by mechanosensing matrix stiffness and myosin activation.

## Discussion

Tamoxifen has been consistently used for many years as an anti-estrogen hormonal therapy for estrogen-positive breast cancers, and the implications of tamoxifen on the nuclear estrogen receptors have been thoroughly described [36]. Here we found that tamoxifen promotes multiple changes in PSCs and the microenvironment of pancreatic cancer through a GPER-dependent mechanism that induces a negative regulation of HIF-1A by decreasing actomyosin-dependent PSC contractility and matrix stiffness mechanosensing (Fig 8). These multiple changes involve upregulation of the number of blood vessels in PDAC tissues.

Our data show that HIF-1A is the unifying factor through which tamoxifen reduces the levels of LOX-L2, MMP-2, and FN in PSCs and PDAC tissues. We note that HIF-1A may act as a converging point to mechanically regulate the adaptive response of PDAC to hypoxia and the overall architecture of the tumor microenvironment. While hypoxia is the most common method of HIF-1A activation, upregulation of HIF-1A expression has also been seen in the presence of oxygen, with G protein-coupled receptors on the cell surface responding to microenvironmental cues [37]. Our observations that tamoxifen can negatively regulate HIF-1A in cancer cells through a hypoxia-independent mechanism open up the possibility to reprogram the mechanosensory machinery in these cells to modulate their proliferation under hypoxic conditions by targeting HIF-1A.

Previous work has demonstrated that treating KPC mice with ATRA, which induces myosin contraction-mediated mechanical quiescence of PSCs [16], reduced hypoxia and increased vascularization [18]. Another study that used the vitamin D receptor to reprogram PSCs to quiescence also showed heightened levels of angiogenesis [38]. PDAC tissues are notoriously known to be poorly vascularized, which hampers drug delivery [1], and reducing stromal stiffness is known to inhibit metastasis and increased drug efficacy in PDAC [17]. Angiotensin inhibitors have also been proposed to alleviate solid stress in pancreatic cancer models by reducing collagen and hyaluronan production, resulting in increased blood vessel density, reduced hypoxia along with improved vascular perfusion, and increased delivery of cytotoxic drugs [6]. Interestingly, a recent study has shown that PDAC metastasis to the liver recapitulated the poorly vascularized milieu characteristic of the primary tumor [39]. Taking into consideration this finding, we propose that by increasing the vascular density, tamoxifen might not only enhance drug delivery to the primary tumor but also to the metastatic microenvironment. However, this remains to be proven by further experiments.

Our findings are consistent with previous studies that have reported decreased levels of MMP-2 in mammary gland tissues of rat treated with tamoxifen [40]. Inhibition of MMP activity by GM6001 (broad spectrum potent MMP inhibitor) in fibroblasts prevented these cells from creating tracks in the ECM for the cancer cells to follow [41]. Thus, tamoxifen may also play a fundamental role in ECM remodeling of PDAC tissues by regulating the levels of MMP-2, the predominant collagenolytic enzyme secreted by PSCs [28]. We also observed that tamoxifen treatment reduced the thickness and alignment of FN fibers. FN is instrumental in the tumor-associated desmoplastic reaction and it has been shown that aligned FN matrices, those that have been remodeled through force-dependent myofibroblast contraction, favor cancer cell invasion [42].

Tamoxifen treatment induced a reduction in the synthesis and secretion by PSCs of collagen, another chief ECM molecule, and changed its content and organization in PDAC tissues. The decrease

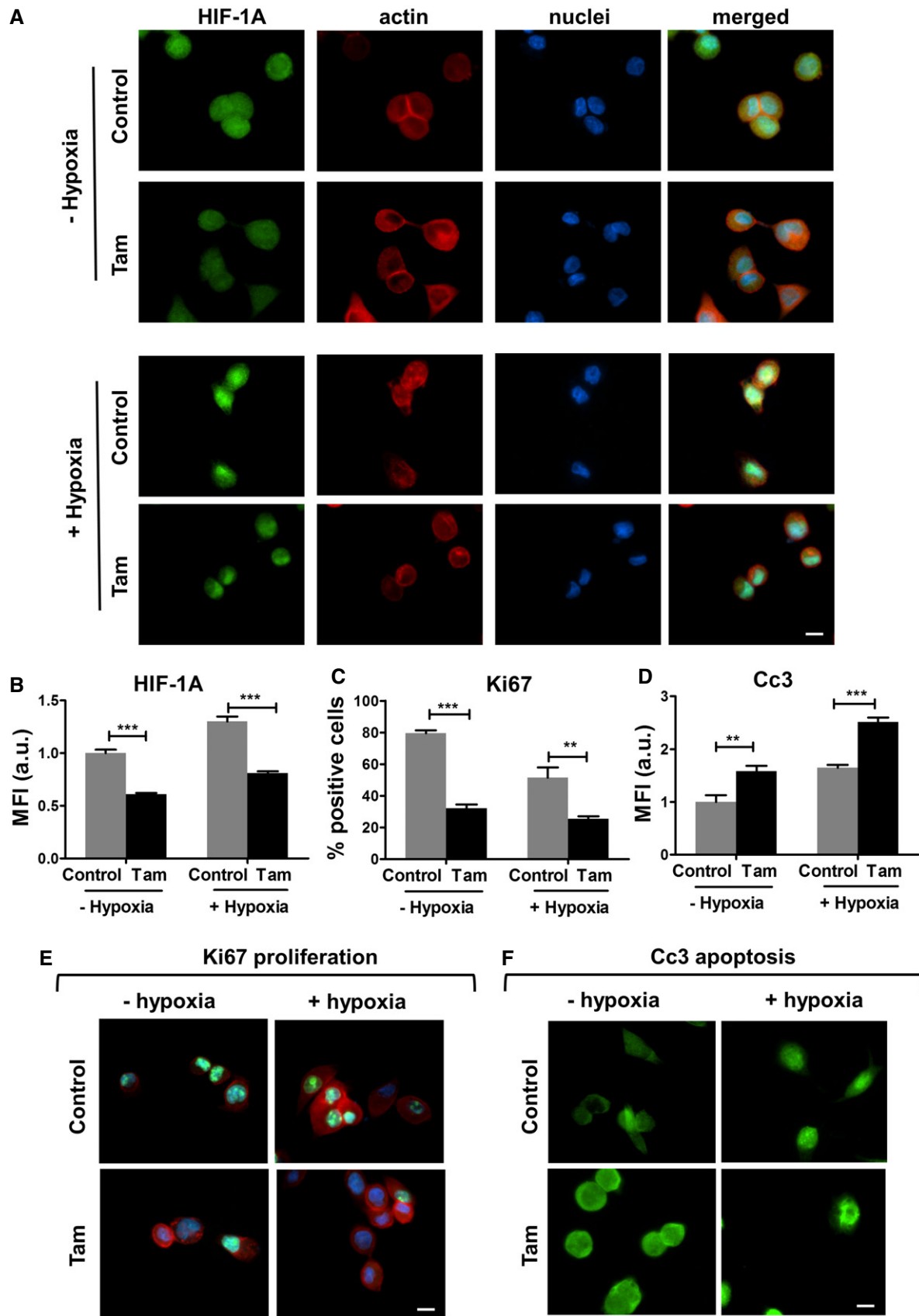

Figure 7.

**Figure 7.  Tamoxifen treatment decreases HIF-1A levels and proliferation and increases apoptosis in Suit-2 pancreatic cancer cells.**

A–F  (A, E, F) Immunofluorescence images of control and tamoxifen-treated Suit2-007 cells, scale bars is 20 μm. Panel (A) represents HIF-1A staining in hypoxia and non-hypoxia conditions, panels (E and F) show Ki67 and Cc3 staining as markers of proliferation and caspase-mediated apoptosis, respectively. Panel (E): red—F-actin, green—Ki67, blue—nuclei. Tamoxifen negatively regulates HIF-1A in hypoxia and non-hypoxia conditions. (B, C, D) Quantification of immunofluorescence staining in panels (A, E, F). For quantification, eight fields of view (*n* > 50 cells) per condition. Histogram bars represent mean ± SEM; **P < 0.01, ***P < 0.001, *t*-test. All panels include data collected during 3 independent experiments.

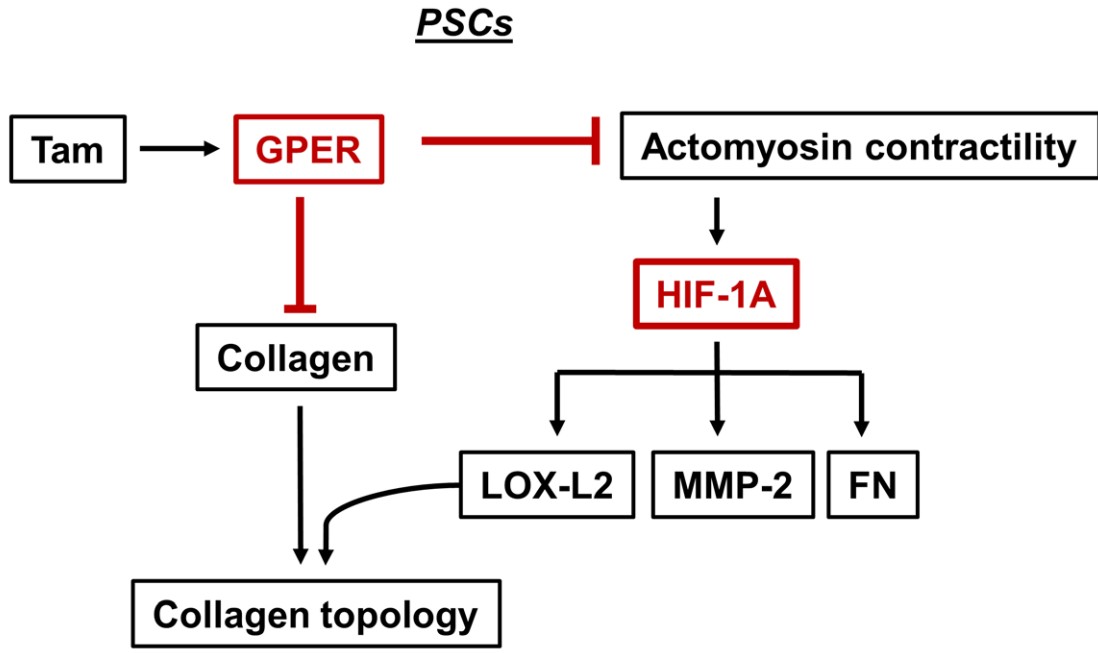

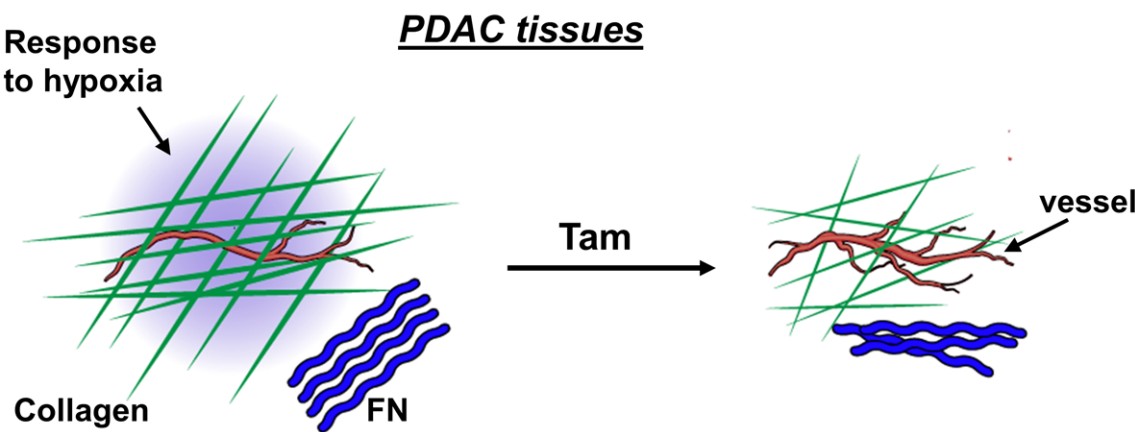

**Figure 8.  Model illustrating the pleiotropic changes of tamoxifen in PSCs and PDAC stroma.**

Top panel: In PSCs, tamoxifen reduces actomyosin contractility via GPER. This mechanically downregulates HIF-1A that is the unifying factor through which tamoxifen reduces LOX-L2, MMP-2, and FN. GPER activation reduces the synthesis and secretion of collagen by PSCs. Tamoxifen also impairs LOX-L2 collagen remodeling. The red color has been used to highlight the two main pillars by which tamoxifen acts on PSCs. Bottom panel: In PDAC tissues, tamoxifen reduces response to hypoxia and increases vascularization, and also reduces the amount and organization of collagen and FN in the ECM.

in the fiber thickness was causally related to the reduction in the LOX-L2 levels, and the decreased alignment could have been promoted by the decreased PSC contractility and tissue tension as others have recently reported [43]. This reduction in the thickness of collagen fibers in the presence of LOX-L2 inhibitor is also in agreement with a previous study that used βAPN to inhibit LOX,

which led to a significant reduction in the fibrillar collagen diameter in mouse tissues [44] and another independent study that used a LOX-L2 antibody to revert ECM remodeling by myofibroblast-secreted LOX-L2 [45]. Increased collagen deposition, alignment, and fiber thickness are promoted by PSCs in PDAC [4,11,13]. Interestingly, two recent studies also found thicker and more aligned collagen fibers adjacent to the PDAC tumors, as well as augmented levels of myosin activation in PDAC tissues from those patients with the shortest survival [4,46]. Taken together, these data underscore collagen topology as a pivotal contributor to the ECM mechanics in PDAC and suggest that treating PSCs with tamoxifen may restore mechanical homeostasis in pancreatic tissues to resolve PDAC-associated fibrosis. It has been recently shown that PDAC cells use collagen fragments as a nutritional source [47]. Therefore, the effect of tamoxifen on decreasing collagen synthesis by PSCs does not only alleviate fibrosis, but also might hamper cancer cells survival under nutrient limited conditions.

In an accompanying report in this issue [48], we show that tamoxifen inhibits the myofibroblastic differentiation of pancreatic stellate cells and their ability to remodel the tumor microenvironment in PDAC via a mechanotransduction mechanism that involves GPER and YAP. Our observations that tamoxifen inhibits fibronectin, collagen, and HIF-1A expression suggest other links between hypoxia and mechanotransduction. Mechanical induction of HIF-1A has been observed in endothelial cells exposed to low shear stress [49], and in the myocardium in response to mechanical stress [50]. The altered ECM generated by HIF-1A activity may promote activation of the transcriptional regulator YAP through mechanotransduction.

Other mechanisms for how HIF-1A activates YAP, and vice versa, have been previously demonstrated. Firstly, HIF-1A upregulates expression of GPRC5A, a GPCR, which then promotes YAP nuclear localization through RhoA [51]. Secondly, HIF-1A and YAP have been shown to colocalize in the nucleus, where YAP facilitates HIF-1A in promoting the upregulation of the *PKM2* gene. PKM2 promotes the switch from oxidative phosphorylation to lactate generation in glycolysis under hypoxic conditions [52,53]. Enhanced glycolysis is a hallmark of PDAC, where the hypoxia induced by tumor growth necessitates the switch from oxidative phosphorylation to hypoxic lactate production [53]. This positive feedback loop suggests the importance of a drug which can inhibit both components. YAP activity has been shown to promote PDAC independently of KRas mutations [54], and hence, this proposed feedback cycle may be integral to this phenomenon. However, since Ras lies upstream of HIF-1A [55], these mechanisms may also be active in KRas-dependent tumors.

GPER-mediated activation of HIF-1A might be more dominant *in vivo* as HIF-1A-mediated gene expression can promote YAP in multiple ways, through GPRC5A and/or through modulation of the extracellular environment. Conversely, YAP is a transcriptional co-activator for HIF-1A to promote glycolysis, and depending on the strength of activation and the importance of glycolysis for PDAC progression, GPER downregulation of YAP activity may also facilitate HIF-1A activity.

Several previous attempts to use anti-angiogenic therapies in PDAC have failed [56–58], and given that the hypovascular environment of PDAC is not directly linked to hypoxia [9], our work lays the groundwork for future questions. GPER agonists could modulate the fibrovascular stroma of PDAC to increase vascular density and perfusion by reducing overall solid stress (through collagen and FN) which would increase intratumoral drug perfusion, while concurrently impeding the adaptive fitness of tumor and stromal cells to survive under hypoxic conditions (via HIF-1A) and thus promoting widespread hypoxic necrosis.

Tamoxifen is a drug with well-established pharmacodynamics and safety. Given the pleiotropic effects of estrogenic signaling, the ubiquitous expression of GPER across different tissues, and the numerous pathways that lie downstream of GPCR signaling, we suggest that tamoxifen may lead a new strategy for drug repurposing to reprogram the tumor microenvironment.

# Materials and Methods

### Mice

KPC mice (Pdx-1 Cre, Kras$^{G12D/+}$, p53$^{R172H/+}$) were randomized to three groups and were injected (IP) with either: (i) vehicle [corn oil], (ii) 2 mg, or (iii) 5 mg of tamoxifen daily starting the same day when PDAC tumor was detected and continuing until mice reached endpoint (for most mice between 8–14 days). After the treatment, mice were sacrificed and pancreatic tissues harvested and used for further analysis. Animals were maintained in conventional animal facilities and monitored daily. All studies were conducted in compliance with the UK Home Office guidelines under license and approved by the local ethical review committee.

### Cell culture and reagents

Primary, cultured-activated human PSCs (passages 6–8, male gender) were purchased from ScienCell Research Laboratories (Carlsbad, USA) and cultured in DMEM/F-12 Ham (Sigma-Aldrich, USA) with 2%FBS (Gibco, Life Technologies, USA), 50 units/ml penicillin and 50 μg/ml streptomycin (Sigma-Aldrich, USA), and 5 ml Fungizone (Gibco, Life Technologies, USA). Tamoxifen (Z-4-hydroxytamoxifen, cat. H7904 Sigma-Aldrich, USA) was prepared in ethanol, and PSCs were treated with 5 μM tamoxifen (or with ethanol vehicle-only) under dim light conditions for 10 days. Media was replenished every 72 h. Cell culture under hypoxic conditions was carried out in anaerobic workstations with oxygen-deprived atmosphere and temperature controlled at 37°C. ER and GPER antagonists were purchased from Tocris and used at 1 μM, ICI182780 (cat. 1047), G15 (cat. 3678). Blebbistatin was from Calbiochem, USA, and UK used at a concentration of 50 μM; human plasma fibronectin (FC010) was from Millipore, USA. siRNA for HIF-1A (cat. Sc-35561) was from Santa Cruz Biotechnology, USA. pEGFP-MRLC1 T18D, S19D (constitutively active MLC-2) was a gift from Tom Egelhoff (Addgene plasmid # 35682). All cell lines were tested for mycoplasma contamination.

### Proteomic analysis

*Sample preparation*
Chemicals were from Sigma-Aldrich unless otherwise mentioned. Water was purified with Milli-Q water purification system (Millipore). The samples were thawed, transferred into tubes with 1 mm

zirconia beads (BioSpec Products), and weighed, after which 100 µl of 8 M urea was added. The samples were homogenized with FastPrep-24 5G homogenizer (MP Biomedicals), followed by sonication and removal of insoluble cell debris by centrifugation. Total protein content of the samples was determined using Thermo Pierce BCA Protein Assay Kit (Thermo Scientific), and the results were measured with CLARIOstar plate reader (BMG Labtech). For further sample preparation, the amount of homogenate equivalent to 100 µg of total protein was taken from each sample in duplicate, followed by dilution to 500 µl with 50 mM ammonium bicarbonate (Fluka Chemie AG). From these dilutions, duplicate pooled QC samples were made, so that the amount of total protein in each sample was 95 µg. Cysteine residues were reduced with dithiothreitol and carbamidomethylated with iodoacetamide. The pH of the samples was adjusted to > 7 with 1 M ammonium bicarbonate, followed by overnight digestion with sequencing grade modified trypsin (Promega). After the digestion, the samples were acidified with 10% trifluoroacetic acid and purified using C18 MicroSpin Columns (The Nest Group Inc.) according to the manufacturer's instructions. LC-MS grade solvents (VWR International) were used. The purified samples were evaporated to dryness in a vacuum centrifuge and stored in −20°C until the LC-MS analysis.

### LC-MS

The samples were resolubilized with sonication in 30 µl of 0.1% trifluoroacetic acid, 1% acetonitrile in LC-MS grade water. Pooled QC samples were used to determine the suitable injection amount (4.0 µl) before the analysis and to monitor system stability throughout the analysis. The LC-MS analysis was done with EASY-nLC 1000 coupled to Q Exactive MS using Xcalibur version 3.1.66.10 (Thermo Scientific). 4.0 µl of tryptic peptide solution, equivalent to approximately 12 µg of total digested protein, was injected into the LC and separated with Thermo Acclaim C18 columns (pre-column: PepMap 100, 75 µm × 2 cm; analytical column: PepMap RSLC, 75 µm × 15 cm). Flow rate was 300 nl/min, and the linear gradients were as follows: 120 min from 5 to 35% of buffer B, followed by 5 min from 35 to 80% of B, 1 min from 80 to 100% of B, and 9 min column wash with 100% of B (A: 1% acetonitrile, 0.1% formic acid, 0.01% trifluoroacetic acid in water; B: 1% water, 0.1% formic acid, 0.01% trifluoroacetic acid in acetonitrile). Mass spectrometry was carried out using top10 data-dependent acquisition, in which the ten most intense ions from one MS1 full scan (resolution 70,000) were subjected to CID and analyzed in MS2 (resolution 17,500). Samples were run in randomized order, with each sample followed by two wash runs, and four pooled QC samples throughout the analysis.

### Data analysis

Pooled QC samples, used to monitor system stability, were assessed first but excluded from the final data analysis. Proteins were identified (1% FDR [false discovery rate] on peptide and protein level) with Andromeda search engine [59], and protein quantification was performed with MaxQuant software [60,61]. The search was done against up-to-date *Mus musculus* UniProtKB/Swiss-Prot proteome with mass tolerance of 6 ppm in MS1 and 20 ppm in MS2, with the maximum of two missed cleavages [62]. LFQ (label-free quantification) intensity was used without further normalization. Contaminant and decoy matches were omitted before the statistical analysis.

To allow statistical testing, missing values were imputed by adding normally distributed noise (mean = non-zero minimum for each protein divided by 100; SD = mean divided by 100) to the LFQ intensity data. Protein abundance comparison was done using 2-tailed unpaired *t*-tests and FDR multiple hypothesis correction. The changes with $P < 0.05$ were considered statistically significant. GO analysis was done with DAVID 6.8 [63, 64].

## RNA sequencing and data analysis

### RNA sequencing

Three biological replicates were prepared for control and treated PSCs. The initial quality check of the control and treated samples was performed using Bioanalyzer and Qubit. RIN values were 10 for the six samples. Sequencing libraries were prepared using a TruSeq Stranded mRNA Library Prep Kit from Illumina. All six samples were pooled together and run on Nextseq500 using 1 × 75 bp read type.

### Data processing

Sequenced reads were trimmed by Trim Galore with "-q 30" option and then mapped to a reference human genome (GRCh38.p10) using STAR [65] with default option. Gene counts data were obtained by quantification from the aligned data using featureCounts [66]. The following genes were used for data normalization (TUBA1A, TUBA1B, TUBA1C, TUBA3D, TUBA3FP, TUBA4A, TUBB, TUBB1, TUBB2A, TUBB2B, TUBB3, TUBB4B, TUBB6, TUBD1, TUBE1, TUBG1, TUBG2). First, RPKM (reads per kilo-base per million) values were calculated from obtained gene counts data, and then, RPKM values for each encoding gene were compared between control and tamoxifen-treated samples using the Mann–Whitney *U*-test (wilcox.exact function in statistical software R). There were no significant differences for these genes in control and tamoxifen-treated PSCs. The smallest obtained *P*-value was 0.1 (Appendix Table S1). Therefore, the gene counts data were normalized using the genes encoding tubulin encoding genes RUVSeq [67]. Subsequently, normalized gene expression data were analyzed by edgeR [68], and genes with false discovery rate < 0.01 were identified as differential expression genes (DEG).

Gene ontology (GO) analysis of DEGs: GO analysis was implemented in R package clusterProfiler [69]. Initially, DEGs were classified into upregulated and downregulated DEGs based on their log2 fold change values. GO enrichment analysis was carried out by *enrichGO* function with the following parameters (ont = "BP", pAdjustMethod = "BH", pvalueCutoff = 0.01, qvalueCutoff = 0.01). To reduce the redundancy of enriched GO terms, *simplify* function was used with the following parameters (cutoff = 0.5, by = "qvalue", select_fun = min). For visualization, *dotplot* function was used with the following parameters (x = "count", showCategory = 15, colorBy = "qvalue").

## RT–PCR

Total RNA was extracted with RNeasy Mini Kit (Qiagen, 74104), and 1 µg of total RNA was reverse transcribed by High-Capacity RNA-to-cDNA Kit (Applied Biosystems, 4387406) according to manufacturer's instructions. qPCR was performed with SYBR Green PCR Master Mix (Applied Biosystems, 4309155) with 100 ng cDNA

input in 20 µl reaction volume. RPLP0 (60S acidic ribosomal protein P0) expression level was used for normalization as a housekeeping gene. The sequences were as following: RPLP0 (F) 5′-CG GTTTCTGATTGGCTAC-3′; RPLP0 (R) 5′-ACGATGTCACTTCCAC G-3′; COL1 (F) 5′-GCTATGATGAGAAATCAACCG-3′; COL1 (R) 5′-TCATCTCCATTCTTTCCAGG-3′; LOX (F) 5′-CAACATTACCACAG TATGGATG-3′; LOX (R) 5′-TAGTCACAGGATGTGTCTTC-3′; LOX-L1 (F) 5′-ATTGTCCAATCCTGATCTCC-3′; LOX-L1 (R) 5′-GAATCCC TGTGGCATC-3′; LOX-L2 (F) 5′-GATGTACAACTGCCACATAG-3′; LOX-L2 (R) 5′-GACAGCTGGTTGTTTAAGAG3′;LOX-L3 (F) 5′-A GGAAACTACATTCTCCAGG-3′; LOX-L3 (R) 5′-ACCCAGATTC TATGTCCATC-3′;LOX-L4 (F) 5′-CTACTACAGGAAAGTCTGGG-3′; LOX-L4 (R) 5′-TCCAGAAGGAGTTCTTATTCG-3′; HIF-1A (F) 5′-AA AATCTCATCCAAGAAGCC-3′; HIF-1A (R) 5′-AATGTTCCAATTCC TACTGC-3′; HIF-2A (F) 5′-CAGAATCACAGAACTGATTGG-3′; HIF-2A (R) 5′-TGACTCTTGGTCATGTTCTC-3′; VEGFA (F) 5′-AATG TGAATGCAGACCAAAG-3′; VEGFA (R) 5′-GACTTATACCGGG ATTTCTTG-3′; VEGFB (F) 5′-GAAAGTGGTGTCATGGATAG-3′; VEGFB (R) 5′-ATGAGCTCCACAGTCAAG-3′; MMP-2 (F) 5′-T CTCCTGACATTGACCTTGGC3′; MMP-2 (R) 5′-CAAGGTGCTG GCTGAGTAGATC3′; MMP-9 (F) 5′-TTGACAGCGACAAGAAGTGG3′; MMP-9 (R) 5′-GCCATTCACGTCGTCCTTAT3′; TIMP-2 (F) 5′-A GAAATATTGGACTTGCTGC-3′; TIMP-2 (R) 5′-GCTTGTCAACTTT CAACAAC-3′. FN1 (F) 5′-CCATAGCTGAGAAGTGTTTTG-3′; FN1 (R) 5′-CAAGTACAATCTACCATCATCC-3′; fibronectin-EDA (F) 5′-TC CAAGCGGAGAGAGT-3′, fibronectin-EDA (R); 5′-GTGGGTGTGAC CTGAG-3′, fibronectin-EDB (F) 5′ -CCACCATTATTGGGTACCGC-3′; fibronectin-EDB (R) 5′-CGCATGGTGTCTGGACCAATG-3′. All primers were used at 300 nM final concentration. The relative gene expression was analyzed by comparative $2^{-\Delta\Delta Ct}$ method.

**Western blotting and zymography**

Cells were washed with chilled PBS and lysed in either Triton X-100 buffer (150 mM sodium chloride, 1% Triton X-100, and 50 mM Tris–HCl, pH 8) or RIPA (Radio Immuno Precipitation Assay) buffer (150 mM sodium chloride, 1% NP-40, 0.5% sodium deoxycholate, 0.1% SDS, and 50 mM Tris–HCl, pH 8) containing 1 mM activated Na3VO4 and protease inhibitors (Complete mini, Roche). Lysate was collected using a cell scraper, disrupted by repetitive trituration through a 25-gauge needle, and incubated for 30 min on ice with periodic mixing. This was followed by centrifugation at 12,000 *g* for 20 min at 4°C. The protein concentration in supernatant was determined using a BCA protein assay kit (Fisher Scientific, UK). Cell lysates were mixed with 4× Laemmli buffer (Bio-Rad) and denatured by heating at 100°C for 5 min. Samples then were loaded into a 4–20% Mini-PROTEAN TGX Precast Gel (Bio-Rad), and proteins were transferred to nitrocellulose membranes (Bio-Rad). Protein on membranes was stained using REVERT total protein stain (LI-COR) as per manufacturer's instructions, and blots were imaged using an Odyssey infrared imaging system. The stain was removed using REVERT Reversal Solution (LI-COR), followed by washing in Tris-buffered saline (TBS). The membranes were blocked in Odyssey blocking buffer (LI-COR) for 1 h followed by overnight incubation with primary antibodies (rabbit anti-collagen type I, ab34710, 1:1,000; rabbit anti-fibronectin, ab2413, 1:1,000 (both Abcam); rabbit anti-MMP-2, sc-10736, 1:500 (Santa Cruz Biotechnology)) in 0.1% Tween in TBS (TBST). After further washes in TBST, blots were incubated

for 1 h with secondary antibodies (Donkey anti-rabbit 680RD or Donkey anti-rabbit 800CW, 1:15,000, LI-COR). Membranes were washed again in TBST and imaged using an Odyssey infrared Imaging system (LI-COR). Total protein for normalization and target protein expression were quantified using Image Studio Lite (Version 5.2, LI-COR). Target protein was normalized to total protein per lane and presented relative to the control group. For LOX-L2, the cell lysates were prepared with radio immunoprecipitation assay (RIPA) buffer (Sigma, R0278) containing proteinase inhibitors (Sigma, P4340). The protein concentration was quantified by DC protein assay (Bio-Rad, 500-0113) according to manufacturer's instructions. Samples were separated by an SDS–PAGE gel under reducing conditions and transferred to a nitrocellulose membrane (GE Healthcare, 10401196), then blocked with 5% bovine serum albumin (BSA, Sigma, A8022)—0.1% Tween-20 (Sigma, P1379) in PBS. LOX-L2 primary antibody (Santa Cruz Biotechnology sc-48724 1/100 diluted) was prepared in blocking solution and incubated overnight at 4°C. The membrane was washed and incubated with horseradish peroxidase (HRP)-conjugated secondary antibodies in blocking solution for 1 h at room temperature. Finally, the membrane was washed and developed with HRP substrate (Millipore, WBLUR0100). For the cases in which the total protein was used as reference.

For the zymography experiments, PSCs were cultured on polyacrylamide gels for 24 h prior to replacing culture medium with serum-free medium, DMEM/Nutrient Mixture F-12 Ham (D8437, Sigma) for a further 24 h. After 24 h, this medium was removed and added to non-reducing Laemmli buffer. For gelatine zymography, recombinant human MMP-2 (PF037, Calbiochem) and recombinant human MMP-9 (PF038, Calbiochem) were prepared as stock solutions of 10 µg/ml and diluted with sterile H2O to a final concentration of 10 and 20 ng/ml, respectively, and added to non-reducing Laemmli buffer. Samples were run and developed according to the protocol reported for zymography by Toth & Fridman [70]. Assayed gels were photographed with a UVP Biospectrum 500 Imaging System. Band digestion intensity representing, potential MMP activity, was calculated using the densitometry plugin for ImageJ.

**Immunohistochemistry**

Formalin-fixed paraffin-embedded (FFPE) blocks were sectioned at 4 µm. Deparaffined and rehydrated with histoclear (National diagnostics, HS-200) followed by decreased concentrations of ethanol, heat-induced antigen retrieval was done by boiling the sections in 10 mM sodium citrate buffer, pH = 6, for 20 min in microwave (for MMP-2 no antigen retrieval was done). Endogenous hydrogen peroxide activity was quenched with 0.3% $H_2O_2$ in Methanol for 30 min at room temperature. Sections were blocked with normal serum for 1 h at room temperature and incubated with primary antibodies diluted in blocking serum overnight at 4°C in a humidified chamber. Primary antibody dilutions are as follows: FN (Abcam, ab2413, 1/250), MMP-2 (Abcam, ab37150, 1/500). Primary antibodies were washed with PBS, and biotinylated secondary antibodies were diluted 1/250 in PBS and incubated for 30 min at room temperature then washed. Sections were incubated with avidin layer (VECTASTAIN Elite ABC Kit, Vector laboratories, PK-6100) for 30 min at room temperature then developed with Peroxidase Substrate (DAB substrate Kit, Vector laboratories, SK-4100). Finally, sections were counterstained with hematoxylin (Abcam, ab128990),

dehydrated by increasing concentration of ethanol and histoclear, and then mounted in DPX mountant (Sigma, 06522). In order to analyze FN staining, the staining was selected from raw images (with hematoxylin) by using IHC toolbox plugin by Fiji. Stain-selected images were thresholded then converted into binary. FN fibril thickness was analyzed on thickness maps which were created with BoneJ plugin on stain-selected images. The alignment score of each image was determined by using custom-made Matlab code. Staining intensities were measured with color deconvolution.

### Immunofluorescence

Cells were fixed with 4% PFA, blocked, and permeabilized with 2% BSA and 0.1% Triton X-100 (all Sigma-Aldrich, St. Louis, MO, USA), and then incubated with primary antibodies (Fibronectin ab2413, Abcam, UK) 1/100 diluted in 2% BSA for 1 h at RT, then washed with PBS and incubated with secondary antibodies (Alexa Fluor® 488 anti-rabbit, Rhodamine Red™-X goat anti-mouse, both Life Technologies, USA) and phalloidin (Alexa Fluor® 546, A22283, Life Technologies, USA) 1/400 diluted in PBS for 45 min in dark. Collagen-I antibody (abcam UK, ab34710 1/100 dilution). HIF-1 alpha antibody (abcam UK, ab2185 1/200 dilution). Ki67 antibody (abcam UK, ab15580 1/200 dilution). Caspase-3 antibody (abcam UK, ab13847 1/100 dilution). Finally the coverslips were mounted with ProLong® Gold Antifade with DAPI (Life Technologies, USA).

For LOX-L2 staining (FFPE), blocks were sectioned, deparaffined, and rehydrated as described previously. Antigen retrieval was done by incubating the sections in boiled citrate buffer pH = 6 for 1 h. LOX-L2 antibody (Santa Cruz Biotechnology sc-66950,1/50) and αSMA antibody (Abcam, UK, ab7817, 1/50), and secondary goat anti-mouse AlexaFluor 488 (Life Technologies, Paisley, UK; A-11029, 1/200) and goat anti-rabbit AlexaFluor 546 (Life Technologies, Paisley, UK; A-11035 1/200) antibodies were used.

For immunofluorescent staining of GLUT1 and CD31 in pancreatic tissues, 6-μm frozen sections were fixed in ice-cold acetone, washed in PBS, and blocked with 10% goat serum (Sigma-Aldrich, St. Louis, MO) and 20 μg/ml AffiniPure Fab goat anti-mouse IgG fragment (Jackson ImmunoResearch Laboratories, Suffolk, UK) for 1 h and 30 min, respectively. Sections were incubated with primary antibodies against GLUT1 (Merk Millipore, UK; 07-1401 1/500) and CD31 (Abcam, Cambridge, UK; ab56299 1/200) in 1% BSA and 0.3% Triton X-100 (both from Sigma-Aldrich, St. Louis, MO) overnight at 4°C. Primary antibodies were washed with 0.1% Triton X-100 and replaced with goat anti-rabbit AlexaFluor 546 (Life Technologies, Paisley, UK; A-11035 1/200) and goat anti-rat AlexaFluor 488 (Life Technologies, Paisley, UK; A11006 1/200) for 1 h at room temperature. Slides were counterstained with Hoechst 33342 (Life Technologies; 1/10,000) and mounted with Prolong Gold (Life Technologies). Sections were imaged with an inverted epifluorescent microscope (Motic AE31 with Moticam 5.0 MP camera) in ≥ 5 200× fields. For GLUT1 stain mean fluorescent intensity, all images were captured using the same exposure settings and processed with ImageJ uniformly across all groups. Image correction for the representative images was similarly kept consistent.

### Micropillar video microscopy and traction force measurements

Pillar arrays were coated with human plasma fibronectin (10 μg/ml; Sigma) and incubated at 37°C for 1 h prior to measurements. Cells that had been trypsinized prior to measurements were suspended in culture media and plated onto the pillar substrates. Time-lapse imaging of the pillars was conducted with an inverted microscope (Eclipse Ti; Nikon) operating in bright-field mode with the samples held at an ambient temperature of 37°C. Image sequences were recorded with an sCMOS camera (Neo sCMOS Andor) at 0.5 Hz using a 40× (0.6NA, air; Nikon) objective over the late spreading phase (90 min < $t$ < 120 min). The position of each pillar in the time-lapse videos was tracked using a custom MATLAB program to track the center of a point spread function of the intensity of the pillars across all frames. By selecting a location free of cells, tracking of a small set of pillars allowed a measurement of the stage drift to be obtained and corrected for in the data set. The time-dependent displacement of a given pillar was obtained by subtracting the initial position of the pillar (zero force) from the position in a given frame. Traction forces were obtained by multiplying the pillar displacements by the pillar stiffness, and the maximum for each pillar was found to obtain the peak forces across the cell.

### 3D ECM remodeling assay

To analyze the ECM remodeling ability of PSCs (control or tamoxifen-treated), Collagen-I (BD Biosciences, 354249, stock concentration 9.37 mg/ml) and Matrigel (BD Biosciences, 354234, stock concentration 9 mg/ml) mixture gels were prepared with 1 part 10× DMEM (Sigma, D2429) and 1 part FBS (Gibco, 10500), yielding to a final concentration of 4.5 mg/ml Collagen-I and 2 mg/ml Matrigel. The gel mixture was neutralized with 1M NaOH (Sigma, S8045), and then, $5 \times 10^5$ cells were embedded in gels in culture media. 80 μl gel volume was added per well of a 96-well plate which was pre-treated with 2% BSA (Sigma, A8022) for 1 h, washed with PBS, and air-dried for 10 min. Gels were set 1 h at 37°C and then incubated with culture media for 3 days at 37°C. For SHG microscopy, gels were prepared as explained above. After 3 days of incubation at 37°C, gels were fixed with 4% paraformaldehyde (PFA) (Sigma, P6148) in PBS for 1 h at 37°C, then washed with PBS and permeabilized with 0.3% Triton X-100 (Sigma, T8787) in PBS for 30 min. After that, gels were blocked with 1% BSA 0.1% Triton X-100 in PBS for 1 h. Gels were washed with PBS and stained with Alexa Fluor 546-conjugated Phalloidin at 1/300 dilution in 1% BSA in PBS for 30 min. Finally, gels were washed two times with PBS.

### Multiphoton microscopy

All SHG images were obtained using a custom-built multiphoton microscope incorporating an upright confocal microscope (SP5, Leica) and a mode-locked Ti:Sapphire Laser (Mai Tai, Newport Spectra-Physics). Images of the SHG signal from collagen-I were collected using an 820 nm excitation with SHG signal obtained with a 414/46 nm bandpass filter and multiphoton autofluorescence signal obtained with a 525/40 nm bandpass filter. A 25×, 0.95 NA water-immersion objective (Leica) was used to deliver the excitation signal and to collect the SHG emission signal from the sample. Images with a 620 × 620 μm field of view were obtained with 2,048 pixel resolution and a line rate of 10 Hz giving a pixel resolution of ~0.3 μm with 3× averaging on each acquisition to reduce the effect of noise.

## Analysis of multiphoton images

SHG images obtained through multiphoton microscopy were analyzed to quantify collagen properties of Matrigels after remodeling. For the assessment of collagen concentration, SHG images obtained for collagen were analyzed in ImageJ for the intensity density across fields of view of 200 μm, taking 5 regions of interest from each full field of view (620 μm). Intensity density values from 35 images were analyzed with the software Prism (GraphPad) for histogram representation. Nonparametric *t*-tests were conducted on the datasets via the Mann–Whitney showing the significant differences between the datasets. SHG images acquired for the imaging of collagen in Matrigels were further analyzed to quantify the orientation of the collagen fibrils. Using the 200 μm field of view images used for intensity density analysis, each image was split into 4 × 100 μm images in ImageJ. Using the fast Fourier transform (FFT) module, all frequencies represented in an image could be represented as a power plot. By fitting an ellipse to the FFT plot, the ratio between the long and short axis of the ellipse could be used to obtain the eccentricity of the plot, where circular behavior (values approaching 1) represents no aligned orientation and lower values represent fiber orientation as alignment is displayed as a power distribution orthogonal to the orientation direction [15]. Eccentricity values were plotted in Prism Software and significance analyzed through the Mann–Whitney nonparametric *t*-test.

## LOX-L2 rescue

To re-introduce LOX-L2 expression in tamoxifen-treated PCSs, PSCs were treated with 5 μM tamoxifen for 10 days, then transfected with 1.5 μg LOX-L2 plasmid (DNASU plasmid repository bank) for 4 h by using JetPRIME reagent (1:3 DNA:jetPRIME ratio (w/v)) and JetPRIME buffer (Polyplus, 114-15). During transfection, cells were cultured in 2 ml media without tamoxifen to exclude the possibility of it affecting the transfection efficiency. After transfection, culture media was changed with 2 ml media containing tamoxifen. Mock transfection was done by using JetPRIME reagent and buffer only (i.e., without DNA), and the cells were otherwise treated the same way as the transfection group. Functional assays were done 48 h after transfection. To study the effect of LOX-L2 over-expression on ECM remodeling, LOX-L2 over-expressing tamoxifen-treated PSCs and mock transfection group were trypsinized and 500,000 cells were embedded in 80 μl Collagen-I Matrigel mixture gels (4.5 and 2 mg/ml final concentration, respectively). After 1-h incubation at 37°C on 2% BSA-treated wells of 96-well plate, gels were covered with tamoxifen containing media and left to be remodeled 3 days at 37°C. In order to block LOX-L2 activity, gels were treated with LOX-L2 blocking antibody during the matrix remodeling process (Santa Cruz Biotechnology sc-48724, Clone:N-15).

## Statistical analysis

All statistical analyses were conducted with the Prism graphical software (GraphPad, Software, La Jolla, CA). Data were generated from multiple repeats of different biological experiments in order to obtain the mean values and standard errors (SEM) displayed throughout. *P*-values have been obtained through *t*-tests on paired or unpaired samples with parametric tests used for data with a normal distribution and nonparametric tests conducted via the Mann–Whitney *U*-test where data had a skewed distribution. Significance for the *t*-tests was set at $P < 0.05$ where graphs show significance through symbols (*$P < 0.05$; **$P < 0.01$; ***$P < 0.001$).

## Data availability

The proteomics analysis data have been deposited in the PeptideAtlas under the reference PASS01070. The PSCs RNA sequencing data have been deposited in the European Nucleotide Archive, accession number ERP023834.

**Expanded View** for this article is available online.

## Acknowledgements

This work was supported by the European Research Council (ERC grant 282051 - ForceRegulation), and the Biotechnology and Biological Sciences Research Council (BBSRC grant BB/N018532/1). M.O.-H. was supported by the International Cooperative Research Program of Institute for Protein Research, Osaka University (ICRa-17-01) and also in part by JSPS KAKENHI Grant No. 15KT0084, Project for Cancer Research and Therapeutic Evolution (P-CREATE), AMED, RIKEN Epigenome and Single Cell Project Grants, Nagase Science Technology Foundation and Astellas Foundation for Research on Metabolic Disorders. TJL was supported by the James Dyson Foundation. We are very grateful to Saadia Karim and Jennifer Morton for providing us with pancreatic tissues from KPC mice and Francesco Di Maggio for help in implementing the initial work with pancreatic stellate cells in the group. We thank Alistair Rice for critical reading of this manuscript.

## Author contributions

EC carried out elastic pillars experiments for traction force measurements and developed the 3D gel remodeling methodology including rescue and functional experiments; BR performed multiphoton microscopy experiments, IF, and image data analysis; EC and MS conducted IHC and IF; TJL carried out hypoxia and angiogenesis studies; EC and DL performed qPCR, IF, and enzymatic assays; SDT performed WB experiments under the supervision of DAL; JST performed proteomics studies under the supervision of MTV; KI conducted analysis of RNA sequencing supervised by MO-H; EC and ADRH conceived the idea for this project and wrote the manuscript with significant inputs from all authors.

## Conflict of interest

The authors declare that they have no conflict of interest.

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
