## [Review Process File · EMBO Reports]

Tamoxifen mechanically reprograms the tumor microenvironment via HIF-1A and reduces cancer cell survival

Ernesto Cortes, Dariusz Lachowski, Benjamin Robinson, Muge Sarper, Jaakko S. Teppo, Stephen D. Thorpe, Tyler J. Lieberthal, Kazunari Iwamoto, David A. Lee, Mariko Okada-Hatakeyama, Markku T. Varjosalo and Armando E. del Río Hernández

Review timeline:	Submission date:	9 June 2018
	Editorial Decision:	16 July 2018
	Revision received:	12 October 2018
	Editorial Decision:	19 October 2018
	Revision received:	19 October 2018
	Accepted:	23 October 2018

Editor: Achim Breiling

Transaction Report: This manuscript was transferred to *EMBO reports* following review at *The EMBO Journal*

1st Editorial Decision

16 July 2018

Thank you for the submission of your manuscript to our editorial offices. We have now received the reports from the referees that were asked to evaluate your study (you will find enclosed below). These were the same four referees that have assessed your previous submission the *The EMBO Journal*.

As you will see, the referees now mostly support the publication of your manuscript in *EMBO reports* (and of the accompanying study submitted back-to-back - see also the decision letter for EMBOR-2018-46556). However, all referees have still several suggestions to improve the paper, and some minor concerns, that we ask you to address in a final revised version of the manuscript. As indicated by referee #1, please discuss your data in the light of the accompanying paper, and cite the other work. Maybe, provide in one of the two papers (I suggest this one) a brief combined discussion/conclusion of both papers. Finally, we do not think that the data of both papers need to be combined (as referee #4 indicates).

I also have the following editorial requests that I ask you to address:

- Please add a short running tile (less than 40 characters w/o spaces) to the title page.
- The Expanded View format, which will be displayed in the main HTML of the paper in a collapsible format, has replaced the Supplementary information. You can submit up to 5 images as Expanded View. Please follow the nomenclature Figure EV1, Figure EV2 etc. The figure legend for these should be included in the main manuscript document file in a section called Expanded View Figure Legends after the main Figure Legends section. Additional Supplementary material should be supplied as a single pdf file labeled Appendix. The Appendix includes a table of content on the first page, page numbers, all figures and their legends. Please follow the nomenclature Appendix Figure Sx throughout the text and also label the figures according to this nomenclature.

For more details please refer to our guide to authors:

<http://embor.embopress.org/authorguide#manuscriptpreparation>

See also our guide for figure preparation:

http://www.embopress.org/sites/default/files/EMBOPress_Figure_Guidelines_061115.pdf

- In Fig. 3A (right panels) there are expanded B&W boxes, but it is unclear where they originate. Please indicate what these are, and where they come from (at least in the figure legend).

- All materials and methods should be included in the main manuscript file.

- Please call out Fig. 5 sequentially in the manuscript text, or change the order of the panels.

- Please call out the different panels of Fig. 6 in the manuscript text.

- It seems there is presently no callout for Fig. S9. Please check.

- You uploaded one excel file of 6 supplementary tables. Please upload the first 5 as EV tables (please use the nomenclature Table EVx throughout the text) and Table 6 as dataset (please use the nomenclature Dataset EV1). Or, include the first 5 tables in the Appendix (Appendix Table S1, S2 ...). Table 6 needs to be a Dataset, though.

- Please indicate in the author checklist in field D10 that you comply with the ARRIVE guidelines, and re-submit the modified form. See also our guidelines for the use of living organisms, and the respective reporting guidelines: <http://embor.embopress.org/authorguide#livingorganisms>

- Please also format the references according to EMBO reports style. See: <http://embor.embopress.org/authorguide#referencesformat>

- We now strongly encourage the publication of original source data with the aim of making primary data more accessible and transparent to the reader. The source data will be published in a separate source data file online along with the accepted manuscript and will be linked to the relevant figure. If you would like to use this opportunity, please submit the source data (for example scans of entire gels or blots, data points of graphs in an excel sheet, additional images, etc.) of your key experiments together with the revised manuscript. Please include size markers for scans of entire gels, label the scans with figure and panel number, and send one PDF file per figure.

- a Microsoft Word file (.doc) of the revised manuscript text
- a letter detailing your responses to the final referee comments in Word format (.doc)
- editable TIFF or EPS-formatted figure files (main figures and EV figures) in high resolution
- the revised Appendix

In addition I would need from you:

- a short, two-sentence summary of the manuscript
- two to three bullet points highlighting the key findings of your study
- a schematic summary figure (in jpeg or tiff format with the exact width of 550 pixels and a height of about 400 pixels) that can be used as visual synopsis on our website.

I look forward to seeing the final revised version of your manuscript when it is ready. Please let me know if you have questions or comments regarding the revision.

REFeree REPORTS

Referee #1:

In the paper 'Tamoxifen mechanically reprograms the tumor microenvironment and the survival of cancer' the authors present data suggesting that tamoxifen treatment of PDAC mice and PSCs in

vitro regulate HIF1a through actomyosin dependent mechanisms that are independent of hypoxia.

The revised manuscript is massively improved from the first version of the paper that this reviewer received from EMBO Journal. The paper is now suitable for publication in EMBO Reports however, this reviewer would appreciate if the authors showed some immunoblotting experiments of HIF1a in figure 2. As all their conclusions in the paper are based on the fact that HIF1a mRNA levels correlates with the HIF1a protein levels, but this is rarely the case with respect to HIF1a as the protein is quickly post-transcriptionally regulated by PHD2 and subsequent ubiquitination-mediated degradation under normoxic conditions. Indeed, their hypoxia experiments (which stabilizes HIF1a proteins) does not entirely rescue the tamoxifen phenotype.

The authors also need to demonstrate that their HIF1a siRNA actually lowers the levels of HIF1a mRNA and protein - no validation is demonstrated.

In several experiments the authors treat cells with tamoxifen (which lower HIF1a) as well as depleting HIF1a with siRNAs. Why do they do this - it gives the same outcome? Instead they should have tried to stabilize HIF1a to rescue the tamoxifen effect - i.e by performing the experiment under hypoxic conditions or by depleting PHD2. This reviewer thinks this does not make sense and actually suggest to remove it entirely.

The two submitted papers submitted back-to-back make a coherent study, which deserve publishing back-to-back in EMBO report if the minor comments are probably answered. However, it would make sense to try to discuss to combined findings in maybe the second paper. For instance, the first paper suggest that GPER regulates YAP signaling while the second paper does not even mention YAP but concentrates on HIF1a signaling.

A combined discussion of the GPER effect on both YAP and HIF signaling would be of value to the reader. What does the authors believe is going on, which of these two transcriptional programs are the most important and are they strictly co-dependent for the progression of PDAC. For instance, are they only important for YAP-induced PDAC which is K-Ras independent?

The two papers deserve a proper discussion. Alone each story is less exiting as compared together. Maybe EMBO Report should consider a 'News and Views'?

Referee #2:

This is an interesting manuscript that reveals how tamoxifen treatment can affect the mechanical properties of pancreatic tumors. The study is generally well performed and thorough and provides significant mechanistic details on how tamoxifen leads to decreased tumor stiffness. The conclusions drawn are well supported by experimental evidence. I have a few minor points that would be good to address.

1. Figure 1A, the control immunofluorescence image looks overexposed.
2. Figure 6 and 7 could use a bit more details in the figure legend.
3. The direct tamoxifen effect on pancreatic cancer cells could be better acknowledged, even if it appears to depend to some degree on the tissue stiffness (that may be attributed to changes in PSCs). I think it should be mentioned in the discussion and perhaps in the abstract.

Referee #3:

Overall, these two submissions have a relatively tight focus on the 'softening effect' of Tamoxifen on PDAC tissues. Conventional expression profiling is used to quantify the loss of contractility and matrix components at protein and transcript levels, giving a consistent picture. Overall, regulatory connections are not always clear,

and some thoroughness seems lacking in the main figures.

In the manuscript "Tamoxifen mechanically reprograms the tumor microenvironment...", my only concerns are:

1. Where are the HIF's in Fig.1E?
- Are the 'Hypoxia genes' all validated targets of HIF-1A (eg. LMNA)?
2. Fig.1E,F normalization by tubulin genes requires showing these are appropriate housekeeping genes.
3. Are these also constant at protein level in Fig.1B,C,H?
4. Blebbistatin should reduce HIF-1A and downstream factors in Fig.8, but evidence in the main figures would help.

Referee #4:

Overall opinion is that the two manuscripts should be combined to make only one paper. None, on its own, is complete. I do not think that a back-to-back publication will help address the issues, since every paper is considered, studied and cited on its own and, therefore, should make sense individually. The authors should make the effort of combining both manuscripts in order to be considered as whole paper for publication in EMBO, or perhaps the authors could prioritize one of the papers.

Reviewer #1's concern: "All of the data provided in the Response (eg. 'Tamoxifen induces PSCs quiescence', Percentage of PDAC Tumor Growth, etc.) needs to be included in Supplement or EV figures. Not only were some of the same questions addressed by other Reviewers (eg. Reviewer#2, question 3.2), but future readers are likely to have similar questions about this study." The authors' answer was: "Many thanks for this comment. The data has been included either in the supplementary information of this manuscript or in the accompanying manuscript (ex-Nat Communication paper) that is proposed to be published back to back in EMBO Reports together with this current manuscript." It is very confusing and time consuming to go back and forth between manuscripts to find where the authors decided to include this data. On top of this, these are very important experiments to prove the mechanism that the authors are describing. Therefore, they should all be included in this manuscript.

-Reviewer #3 also addressed this concern: "Functional results on TAM mediated molecular changes would significantly strengthen the study. In the response to reviewer, the authors mention that functional data are included in a previously submitted accompanied manuscript. However, this may not help much to improve the completeness of this current manuscript. Is a common functional role of the molecular mechanisms a unifying factor?"

1st Revision - authors' response

12 October 2018

Referee #1:

In the paper 'Tamoxifen mechanically reprograms the tumor microenvironment and the survival of cancer' the authors present data suggesting that tamoxifen treatment of PDAC mice and PSCs in vitro regulate HIF1a through actomyosin dependent mechanisms that are independent of hypoxia.

The revised manuscript is massively improved from the first version of the paper that this reviewer received from EMBO Journal. The paper is now suitable for publication in EMBO Reports however, this reviewer would appreciate if the authors showed some immunoblotting experiments of HIF1a in figure 2. As all their conclusions in the paper are based on the fact that HIF1a mRNA levels correlates with the HIF1a protein levels, but this is rarely the case with respect to HIF1a as the protein is quickly post-transcriptionally regulated by PHD2 and subsequent ubiquitination-mediated degradation under normoxic conditions. Indeed, their hypoxia experiments (which stabilizes HIF1a proteins) does not entirely rescue the tamoxifen phenotype.

Authors: Many thanks for this insight. We used Western blot to determine the protein levels of HIF-1A in control and tamoxifen treated pancreatic stellate cells (PSCs).

In the main text: We used immunoblotting to investigate the effect of tamoxifen on the levels of HIF-1A in PSCs and observed an overall 25% reduction in the 3 main HIF-1A isoforms (1-3) and also in the posttranslational modified HIF-1A (Figure 1J and Appendix Figure S3).

The authors also need to demonstrate that their HIF1a siRNA actually lowers the levels of HIF1a mRNA and protein - no validation is demonstrated.

Authors: We verified the efficiency of the siRNA HIF1A in lowering HIF1A in PSCs at the protein level (using IF) and mRNA level (using PCR). These results are presented in Appendix Figure S4.

In several experiments the authors treat cells with tamoxifen (which lower HIF1a) as well as depleting HIF1a with siRNAs. Why do they do this - it gives the same outcome? Instead they should have tried to stabilize HIF1a to rescue the tamoxifen effect - i.e by performing the experiment under hypoxic conditions or by depleting PHD2. This reviewer thinks this does not make sense and actually suggest to remove it entirely.

Authors: HIF-1A has been shown to control the expression and activation of the members of the LOX family, MMP-2 and fibronectin. The comparison between the tamoxifen group and the tamoxifen group in which HIF-1A was knocked down was used to investigate if the effect of tamoxifen in LOX-L2, MMP-2, and fibronectin was mediated by HIF-1A or if there was another concurrent mechanism. The fact that we observed no significant differences in cells treated with tamoxifen (with and without HIF-1A depletion) suggests that the effect of tamoxifen in these 3 targets should be mediated by HIF-1A.

The two submitted papers submitted back-to-back make a coherent study, which deserve publishing back-to-back in EMBO report if the minor comments are probably answered. However, it would make sense to try to discuss to combined findings in maybe the second paper. For instance, the first paper suggest that GPER regulates YAP signaling while the second paper does not even mention YAP but concentrates on HIF1a signaling.

A combined discussion of the GPER effect on both YAP and HIF signaling would be of value to the reader. What does the authors believe is going on, which of these two transcriptional programs are the most important and are they strictly co-dependent for the progression of PDAC. For instance, are they only important for YAP-induced PDAC which is K-Ras independent? The two papers deserve a proper discussion. Alone each story is less exiting as compared together. Maybe EMBO Report should consider a 'News and Views'?

Authors: We included in this manuscript a discussion of the results from both papers and the potential interconnection between the GPER/YAP and GPER/HIF-1A axes.

In the main text (discussion section): Ref 47 below is the accompanying manuscript

In an accompanying report in this issue (47), we show that tamoxifen inhibits the myofibroblastic differentiation of pancreatic stellate cells and their ability to remodel the tumor microenvironment in PDAC via a mechanotransduction mechanism that involves GPER and YAP. Our observations that tamoxifen inhibits fibronectin, collagen and HIF-1A expression suggests other links between hypoxia and mechanotransduction. Mechanical induction of HIF-1A has been observed in endothelial cells exposed to low shear stress (48), and in the myocardium in response to mechanical stress (49). The altered ECM generated by HIF-1A activity may promote activation of the transcriptional regulator YAP through mechanotransduction.

Other mechanisms for how HIF-1A activates YAP, and vice versa, have been previously demonstrated. Firstly, HIF-1A upregulates expression of GPRC5A, a GPCR, which then promotes YAP nuclear localisation through RhoA (50). Secondly, HIF-1A and YAP have been shown to colocalise in the nucleus, where YAP facilitates HIF-1A in promoting the upregulation of the PKM2 gene. PKM2 promotes the switch from oxidative phosphorylation to lactate generation in glycolysis under hypoxic conditions (51, 52). Enhanced glycolysis is a hallmark of PDAC, where the hypoxia

induced by tumour growth necessitates the switch from oxidative phosphorylation to hypoxic lactate production (52). This positive feedback loop suggests the importance of a drug which can inhibit both components. YAP activity has been shown to promote PDAC independently of KRas mutations (53), and hence this proposed feedback cycle may be integral to this phenomenon. However, since Ras lies upstream of HIF-1A (54), these mechanisms may also be active in KRas-dependent tumors.

GPER mediated activation of HIF-1A might be more dominant in vivo as HIF-1A mediated gene expression can promote YAP in multiple ways, through GPRC5A and/or through modulation of the extracellular environment. Conversely, YAP is a transcriptional co-activator for HIF-1A to promote glycolysis, and depending on the strength of activation and the importance of glycolysis for PDAC progression, GPER downregulation of YAP activity may also facilitate HIF-1A activity.

Referee #2:

This is an interesting manuscript that reveals how tamoxifen treatment can affect the mechanical properties of pancreatic tumors. The study is generally well performed and thorough and provides significant mechanistic details on how tamoxifen leads to decreased tumor stiffness. The conclusions drawn are well supported by experimental evidence. I have a few minor points that would be good to address.

1. Figure 1A, the control immunofluorescence image looks overexposed.

Authors: Many thanks for spotting this. This figure has been changed adding a new image that better represents the immunofluorescence staining for this condition.

2. Figure 6 and 7 could use a bit more details in the figure legend.

Authors: More details have been added to these figures following this reviewer's recommendation. The added text is underlined.

Initially submitted:

Figure 6: Tamoxifen treatment decreases proliferation and increases apoptosis in epithelial cells of PDAC tissues. (A) Immunofluorescence images of PDAC tissues from KPC mice treated with vehicle control, and 2mg of tamoxifen, scale bar is 100 mm. White arrows indicate Ki67 positive nuclei. (B) Quantification of staining in panel A (n=4 animals per condition, and n \geq 5 sections per animal). Histograms bars represent mean \pm s.e.m.; ***P<0.001, t-test.

Figure 7: Tamoxifen treatment decreases HIF-1A and proliferation and increases apoptosis in Suit-2 pancreatic cancer cells. (A, E, F) Immunofluorescence images of control and tamoxifen treated suit-2 cells, scale bar is 20 mm. (B,C,D) Quantification of immunofluorescence staining in panels A, E, F. For quantification, eight fields of view (n>50 cells) per condition. Histograms bars represent mean \pm s.e.m.; ***P<0.001, t-test.

Current version:

Figure 6: Tamoxifen treatment decreases proliferation and increases apoptosis in epithelial cells of PDAC tissues. (A) Immunofluorescence images of PDAC tissues from KPC mice treated with vehicle control, and 2mg of tamoxifen, scale bar is 100 mm. Upper panels: Ki67 staining is used as a surrogate of proliferation. White arrows indicate Ki67 positive nuclei in epithelial cells. Lower panels: Cc3 staining shows the cells undergoing caspase-mediated apoptosis. Tamoxifen panels show higher levels of yellow staining, which indicates higher percentage of apoptotic epithelial cells. (B) Quantification of staining in panel A (n=4 animals per condition, and n \geq 5 sections per animal, two experimental repetitions). Histograms bars represent mean \pm s.e.m.; ***P<0.001, t-test.

Figure 7: Tamoxifen treatment decreases HIF-1A levels and proliferation and increases apoptosis in Suit-2 pancreatic cancer cells. (A, E, F) Immunofluorescence images of control and tamoxifen treated suit-2 cells, scale bar is 20 mm. Panel A represents HIF-1A staining in hypoxia and non-hypoxia conditions, panels E and F show Ki67 and Cc3 staining as markers of proliferation and caspase-mediated apoptosis, respectively. Panel E: Red – F-actin, green – Ki67, blue – nuclei.

Tamoxifen negatively regulates HIF-1A in hypoxia and non-hypoxia conditions. (B,C,D) Quantification of immunofluorescence staining in panels A, E, F. For quantification, eight fields of view (n>50 cells) per condition. Histogram bars represent mean \pm s.e.m.; ***P<0.001, t-test. All panels include data collected during 3 independent experiments.

3. The direct tamoxifen effect on pancreatic cancer cells could be better acknowledged, even if it appears to depend to some degree on the tissue stiffness (that may be attributed to changes in PSCs). I think it should be mentioned in the discussion and perhaps in the abstract.

Authors: Many thanks for this comment. The direct tamoxifen effect on pancreatic cancer cells is acknowledged in the discussion section of the revised text as follows:

***In the manuscript:** Our observations that tamoxifen can negatively regulate HIF-1A in cancer cells through a hypoxia independent mechanism open up the possibility to reprogram the mechanosensory machinery in these cells to modulate their proliferation under hypoxic conditions by targeting HIF-1A.*

....GPER agonists could modulate the fibrovascular stroma of PDAC to increase vascular density and perfusion by reducing overall solid stress (through collagen and FN) which would increase intratumoral drug perfusion, while concurrently impeding the adaptive fitness of tumor and stromal cells to survive under hypoxic conditions (via HIF-1A) and thus promoting widespread hypoxic necrosis.

Referee #3:

Overall, these two submissions have a relatively tight focus on the 'softening effect' of Tamoxifen on PDAC tissues. Conventional expression profiling is used to quantify the loss of contractility and matrix components at protein and transcript levels, giving a consistent picture. Overall, regulatory connections are not always clear, and some thoroughness seems lacking in the main figures.

In the manuscript "Tamoxifen mechanically reprograms the tumor microenvironment...", my only concerns are:

1. Where are the HIF's in Fig.1E?

Authors: Because of the relevance of the HIF and VEGF families, we presented this data separately in Appendix Figure S1.

***In the manuscript:** We then focused on the hypoxia inducible factor (HIF) and vascular endothelial growth factor (VEGF) families as key players associated with hypoxia and blood vessels respectively (Appendix Figure S1). For the HIF family, we found that while HIF-3A did not change, HIF-1A was significantly reduced and HIF-2A (also known as EPAS1) was significantly upregulated. We observed an increase in the expression levels of VEGFB and a downregulation of VEGFC. qPCR was used to validate RNA sequencing data that showed a clear trend but did not display significant differences.*

Are the 'Hypoxia genes' all validated targets of HIF-1A (eg. LMNA)?

Authors: The genes reported in the literature to be validated target of HIF-1A are presented in Appendix Figure S2. A previous study reported that mutation in the residue S143P in LMNA affects the transcription of the HIF-1A gene (PMID: 27235420).

2. Fig.1E,F normalization by tubulin genes requires showing these are appropriate housekeeping genes.

Authors: The Appendix Table S1 shows the comparison of the levels of the tubulin genes used for normalization. The smallest p value was 0.1. There were no significant differences between the control and tamoxifen treated groups for these genes.

In the manuscript – methods section: The following genes were used for data normalization (TUBA1A, TUBA1B, TUBA1C, TUBA3D, TUBA3FP, TUBA4A, TUBB, TUBB1, TUBB2A, TUBB2B, TUBB3, TUBB4B, TUBB6, TUBD1, TUBE1, TUBG1, TUBG2). First, RPKM (reads per kilo-base per million) values were calculated from obtained gene counts data, and then, RPKM values for each encoding gene were compared between control and tamoxifen-treated samples using the Mann–Whitney test (wilcox.exact function in statistical software R). There were no significant differences for these genes in control and tamoxifen treated PSCs. The smallest p value was 0.1 (Appendix Table S1).

3. Are these also constant at protein level in Fig.1B,C,H?

Authors: In Fig 1B, C, H, no normalization using the tubulin genes were made. The panels 1B,H represent the immunostaining quantification for Glut1 and HIF-1A, respectively. These intensities were normalized to the control group (no normalization using the tubulin genes were done). The measurements are for tissues coming from PDAC tissues. The data presented in panels E-F are RNA sequencing for pancreatic stellate cells (where the normalization using tubulin genes was done).

For the proteomic data in panel 1C: The normalized intensity of each protein is compared against its own value in control. The proteomics data is not normalized against any “housekeeping” proteins, but instead with the MaxLFQ algorithm in the MaxQuant software.

The normalization used in the proteomic shotgun analysis is described in the paper (<https://www.ncbi.nlm.nih.gov/pubmed/24942700>), but briefly, it assumes that the majority of proteins does not vary between sample groups (a valid assumption for studies such as this, where the same tissue is sampled under different treatment conditions) and thus uses all proteins’ average behaviour as a reference for normalization.

4. Blebbistatin should reduce HIF-1A and downstream factors in Fig.8, but evidence in the main figures would help.

Authors: We agree with this Reviewer that blebbistatin should reduce HIF-1A. We present experimental evidence to support this in Appendix Figure S7. BBI is blebbistatin

Referee #4:

Overall opinion is that the two manuscripts should be combined to make only one paper. None, on its own, is complete. I do not think that a back-to-back publication will help address the issues, since every paper is considered, studied and cited on its own and, therefore, should make sense individually. The authors should make the effort of combining both manuscripts in order to be considered as whole paper for publication in EMBO, or perhaps the authors could prioritize one of the papers.

Reviewer #1's concern: "All of the data provided in the Response (eg. 'Tamoxifen induces PSCs quiescence', Percentage of PDAC Tumor Growth, etc.) needs to be included in Supplement or EV figures. Not only were some of the same questions addressed by other Reviewers (eg. Reviewer#2, question 3.2), but future readers are likely to have similar questions about this study." The authors' answer was: "Many thanks for this comment. The data has been included either in the supplementary information of this manuscript or in the accompanying manuscript (ex-Nat Communication paper) that is proposed to be published back to back in EMBO Reports together with this current manuscript." It is very confusing and time consuming to go back and forth between manuscripts to find where the authors decided to include this data. On top of this, these are very important experiments to prove the mechanism that the authors are describing. Therefore, they should all be included in this manuscript.

-Reviewer #3 also addressed this concern: "Functional results on TAM mediated molecular changes would significantly strengthen the study. In the response to reviewer, the authors mention that

functional data are included in a previously submitted accompanied manuscript. However, this may not help much to improve the completeness of this current manuscript. Is a common functional role of the molecular mechanisms a unifying factor?"

Authors: We have changed manuscript 1 (EMBOR-2018-46556V1) to a short communication format and included a reference to the accompanying manuscript in the results and discussion section. We think this improved the flow of the manuscript 1.

In the main text of EMBOR-2018-46556V1: Ref 54 is the accompanying manuscript

Interestingly, in our study cancer cell invasion was assessed in matrices, which were deprived of both the remodeling PSCs and any soluble factor that might allow PSC–cancer cell communication. This suggests that under tamoxifen treatment PSCs are unable to biomechanically and topologically remodel the ECM, and the resulting ECM architecture is not as conducive to cancer cell invasion. This idea is consistent with previous works that have shown that increased ECM stiffness can promote cancer cell invasion [43] and that RhoA-mediated fibroblast ECM remodeling enhances invasion [51, 52]. Also, increased alignment & thickness of the collagen fibers enhance cancer cell invasion [53]. Tamoxifen also reduces the expression of the cross-linking enzyme LOX–L2 and the matrix metalloproteinase MMP-2 in PSCs [54], which suggests that it is not only the ECM rigidity what influences cancer cell invasion, but that other topological factors play an important role, such as the presence of tracks promoted by MMPs [51] and the alignment and thickness of collagen fibers [53].

Furthermore, in the accompanying manuscript 2, EMBOR-2018-46557V1, we included a discussion about the possible interconnection of the two main mechanisms we presented in both works (GPER/YAP and GPER/HIF1A axes). We also referenced the accompanying manuscript in EMBOR-2018-46557V1. We hope that this will add clarity to the messages conveyed in both manuscripts and make a comprehensive and coherent story.

In the main text of EMBOR-2018-46557V1: Ref 47 is the accompanying manuscript

In an accompanying report in this issue (47), we show that tamoxifen inhibits the myofibroblastic differentiation of pancreatic stellate cells and their ability to remodel the tumor microenvironment in PDAC via a mechanotransduction mechanism that involves GPER and YAP. Our observations that tamoxifen inhibits fibronectin, collagen and HIF-1A expression suggests other links between hypoxia and mechanotransduction. Mechanical induction of HIF-1A has been observed in endothelial cells exposed to low shear stress (48), and in the myocardium in response to mechanical stress (49). The altered ECM generated by HIF-1A activity may promote activation of the transcriptional regulator YAP through mechanotransduction.

Other mechanisms for how HIF-1A activates YAP, and vice versa, have been previously demonstrated. Firstly, HIF-1A upregulates expression of GPRC5A, a GPCR, which then promotes YAP nuclear localisation through RhoA (50). Secondly, HIF-1A and YAP have been shown to colocalise in the nucleus, where YAP facilitates HIF-1A in promoting the upregulation of the PKM2 gene. PKM2 promotes the switch from oxidative phosphorylation to lactate generation in glycolysis under hypoxic conditions (51, 52). Enhanced glycolysis is a hallmark of PDAC, where the hypoxia induced by tumour growth necessitates the switch from oxidative phosphorylation to hypoxic lactate production (52). This positive feedback loop suggests the importance of a drug which can inhibit both components. YAP activity has been shown to promote PDAC independently of KRas mutations (53), and hence this proposed feedback cycle may be integral to this phenomenon. However, since Ras lies upstream of HIF-1A (54), these mechanisms may also be active in KRas-dependent tumors.

GPER mediated activation of HIF-1A might be more dominant in vivo as HIF-1A mediated gene expression can promote YAP in multiple ways, through GPRC5A and/or through modulation of the extracellular environment. Conversely, YAP is a transcriptional co-activator for HIF-1A to promote glycolysis, and depending on the strength of activation and the importance of glycolysis for PDAC progression, GPER downregulation of YAP activity may also facilitate HIF-1A activity.

2nd Editorial Decision

19 October 2018

Thank you for the submission of your revised manuscript to our editorial offices. We think that the remaining points of the referees have been sufficiently addressed.

However, the following editorial requests remain to be addressed before we can proceed with formal acceptance.

- It seems the Appendix file is missing from the V2 submission. Please provide the appendix file. Please make sure that the Appendix includes a TOC with page numbers.

- I suggest to move the EV tables 1-5 into the Appendix file. I do not think it is necessary that these tables are directly shown in the online version of the paper. It would be sufficient if interested readers can refer to the Appendix to find this information. Please move these into the Appendix. Please follow the nomenclature Appendix Table Sx, and update the callouts in the manuscript text accordingly.

- Please call out the different panels of Fig. 6 in the manuscript text.

- Please also format the references according to EMBO reports style (similar to the accompanying manuscript). Please use 'et al' if there are more than ten authors, but the first ten authors need to be shown. Further, references need to be marked by square brackets in the manuscript text. See also: <http://embor.embopress.org/authorguide#referencesformat>

- Please provide a schematic summary figure (in jpeg or tiff format with the exact width of 550 pixels and a height of not more than 400 pixels) that can be used as a visual synopsis on our website.

I look forward to seeing the final revised version of your manuscript when it is ready. Please let me know if you have questions or comments regarding the revision.

2nd Revision - authors' response

19 October 2018

The authors performed all minor editorial changes.

Corresponding Author Name: Armando E. del Río Hernández

Manuscript Number: EMBOR-2018-46557V1